# Hyperspectral Image Fusion with Spectral-Band and Fusion-Scale Agnosticism

**Yu-Jie Liang** [1]  **Zihan Cao** [1]  **Liang-Jian Deng** [1]  **Yang Yang** [1]  **Malu Zhang** [1]

## Abstract

Current deep learning models for Multispectral and Hyperspectral Image Fusion (MS/HS fusion) are typically designed for fixed spectral bands and spatial scales, which limits their transferability across diverse sensors. To address this, we propose SSA, a unified framework for MS/HS fusion with spectral-band and fusion-scale agnosticism. Specifically, we introduce Matryoshka Kernel (MK), a novel operator that enables a single model to process varying ordered spectral band counts within a supported range. Meanwhile, we build SSA upon an Implicit Neural Representation (INR) backbone that models the HS signal as a continuous function, enabling reconstruction at arbitrary spatial resolutions. Together, these two mechanisms support a single MS/HS fusion model for heterogeneous sensors and arbitrary query scales. Extensive experiments demonstrate that our single jointly trained model achieves state-of-the-art performance, generalizes to unseen spatial scales, and transfers to unseen sensors with few-step adaptation. The code can be obtained at https://github.com/vg219/SSA.

## 1. Introduction

Hyperspectral imaging (HSI) (Amigo, 2019) provides rich spectral information across hundreds of narrow bands, making it invaluable in domains like medical diagnostics (Fauvel et al., 2013), remote sensing, and target detection (Uzair et al., 2013; Cheng et al., 2025). However, a fundamental trade-off in optical systems means this rich spectral detail often comes at the cost of low spatial resolution, hampering downstream tasks such as classification (He et al., 2017; Zeng et al., 2024) and object tracking (Chen et al., 2021a; Wu et al., 2024; Jin et al., 2024). To overcome this,

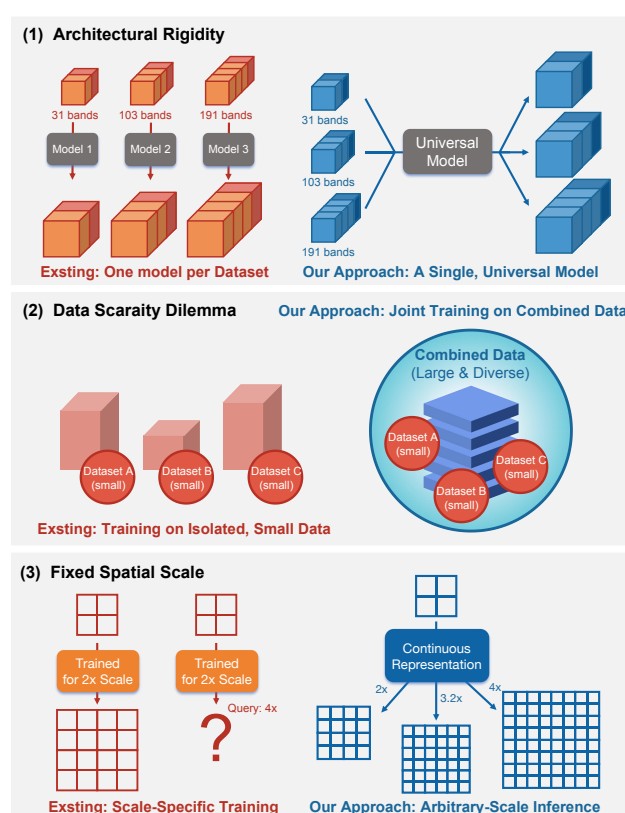

*Figure 1.* Existing paradigms *v.s.* our universal approach for hyperspectral image fusion. Our method can suit different spectral bands joint training and can inference on arbitrary-scale fusion.

Multispectral and Hyperspectral Image Fusion (MS/HS Fusion) has emerged as a key solution, aiming to reconstruct a high-resolution HSI by fusing a low-resolution HSI with a high-resolution multispectral image of the same scene. In recent years, with the rapid development of deep learning, DL-based methods (Masi et al., 2016; Yang et al., 2017; Dian et al., 2018) have gradually become the mainstream for MS/HS Fusion. These methods can automatically learn complex nonlinear relationships, exhibiting immense potential in HSI fusion tasks.

Despite this significant progress, the real-world applicability of DL-based methods is hampered by the deep-rooted challenge of *sensor diversity* (Hong et al., 2025). Datasets captured by different sensors, such as AVIRIS (224 bands) or Pavia University (103 bands), vary drastically in their number of spectral bands, which exposes a fundamental

---

[1]University of Electronic Science and Technology of China, Chengdu, China. Correspondence to: Liang-Jian Deng, Malu Zhang <liangjian.deng@uestc.edu.cn, maluzhang@uestc.edu.cn>.

*Proceedings of the 43$^{rd}$ International Conference on Machine Learning*, Seoul, South Korea. PMLR 306, 2026. Copyright 2026 by the author(s).

lack of generality in current models. As illustrated in Fig. 1, this limitation manifests in three primary ways: *(1) Architectural Rigidity*: Most deep neural networks are designed with hard-coded channel dimensions, forcing researchers to either train independent models for each sensor (Deng et al., 2023b; Wu et al., 2025) or resort to costly fine-tuning (Gonzalez et al., 2025). Some even manually select band subsets (Suryanarayana et al., 2025), which sacrifices valuable spectral information. *(2) The Dilemma of Data Scarcity and Model Scale*: HSI datasets are typically small. Training on such isolated, small-scale data constrains model capacity and hinders the learning of generalizable features, leading to a high risk of overfitting (Hoffmann et al., 2022; Yan et al., 2025; Wang et al., 2026). *(3) Fixed Spatial Scaling Factor*: Current models (Deng et al., 2023a; Hu et al., 2022) are trained for a fixed integer upscaling factor (*e.g.*, $4\times$, $8\times$). They learn a discrete grid-to-grid mapping, failing to handle the arbitrary or non-integer scales required in many applications.

To dismantle these barriers, we introduce SSA, a unified framework that addresses the above challenges through two technical innovations. To tackle spectral rigidity (1) and its impact on data scarcity (2), since hard-coded architectures prevent pooling data across sensors for large-scale training, we propose the Matryoshka Kernel (MK). Inspired by Matryoshka Representation Learning (MRL) (Kusupati et al., 2022), MK allows a single model to process varying ordered spectral band counts, enabling joint training on heterogeneous datasets. Concurrently, to resolve spatial rigidity (3), we build our framework upon an Implicit Neural Representation (INR) backbone, which models signals as continuous functions and supports reconstruction at arbitrary (including non-integer) scaling factors. In this work, we integrate these concepts into a cohesive MS/HS fusion model as a single unified framework. Our contributions are threefold:

- We introduce Matryoshka Kernels (MK), a novel architectural principle realized through adaptive convolution layers. This allows our model to natively process varying ordered spectral band counts within a supported range, enabling spectral-band agnosticism.

- By building our framework upon an Implicit Neural Representation (INR), we replace discrete scaling with a continuous function, enabling reconstruction at arbitrary spatial scales. The unified framework combines variable-band spectral handling and arbitrary-scale decoding in one end-to-end model.

- Extensive experiments show that SSA achieves state-of-the-art performance while generalizing well to unseen scaling factors and transferring to unseen sensors with few-step adaptation, substantially reducing the need for per-sensor retraining.

## 2. Related Works

### 2.1. Deep Learning for MS/HS Image Fusion

Following PanNet (Yang et al., 2017), which introduced deep learning to hyperspectral and multispectral (MS/HS) image fusion, the research paradigm has shifted from traditional methods, such as matrix factorization (Kawakami et al., 2011; Huang et al., 2013) and sparse coding (Nezhad et al., 2016), to data-driven deep learning. Current research largely focuses on designing novel network architectures, for instance, by incorporating attention mechanisms and Transformers (Fang et al., 2024; Wu et al., 2025; Cao et al., 2025) or adopting popular paradigms like GANs (Shang et al., 2024; Zhou et al., 2022) and diffusion models (Cao et al., 2024). However, these highly specialized architectures are often deeply coupled with specific data specifications (e.g., the number of spectral bands), leading to a severe lack of universality and transferability.

Recently, several works (Zhang et al., 2022; Chen et al., 2023) have begun to apply Implicit Neural Representation (INR) to HSI data, modeling the image as a continuous function of coordinates. This approach enables reconstruction at arbitrary scales, neatly addressing the spatial rigidity of conventional models. Subsequent research has explored various extensions, such as integrating low-rank properties (Wang et al., 2025b), performing fusion in the frequency domain (Liang et al., 2024), or using adaptive sampling (Deng et al., 2025). However, despite solving spatial rigidity, these methods face two key limitations: First, they lack spectral flexibility, as models must be re-engineered for datasets with different numbers of bands. Second, as data-driven approaches, achieving strong generalization can be difficult given the limited volume of available HSI data.

### 2.2. Matryoshka Representation Learning and Flexible Model Design

Our solution is philosophically inspired by the broader trend in machine learning towards building more adaptive and general-purpose models. From the perspective of representation granularity adaptation, Matryoshka Representation Learning (MRL) (Kusupati et al., 2022; Hojjat et al., 2025; Cappellazzo et al., 2025) was proposed to explore model flexibility. The core idea of MRL is to train a single, high-dimensional feature vector $z \in \mathbb{R}^d$ that encodes information at different granularities in a nested fashion, such that its shorter prefixes also serve as valid representations. This paradigm provides flexible representations for downstream tasks with varying computational demands.

Formally, a nested representation $z$ obtained by a deep neural network $z := F(\cdot, \theta_F) : \mathcal{X} \to \mathbb{R}^d$ contains different granularities representations: $z_{1:m} \in \mathbb{R}^m$, where $m$ is the desired dimension and $(r : s)$ means one sub-representation

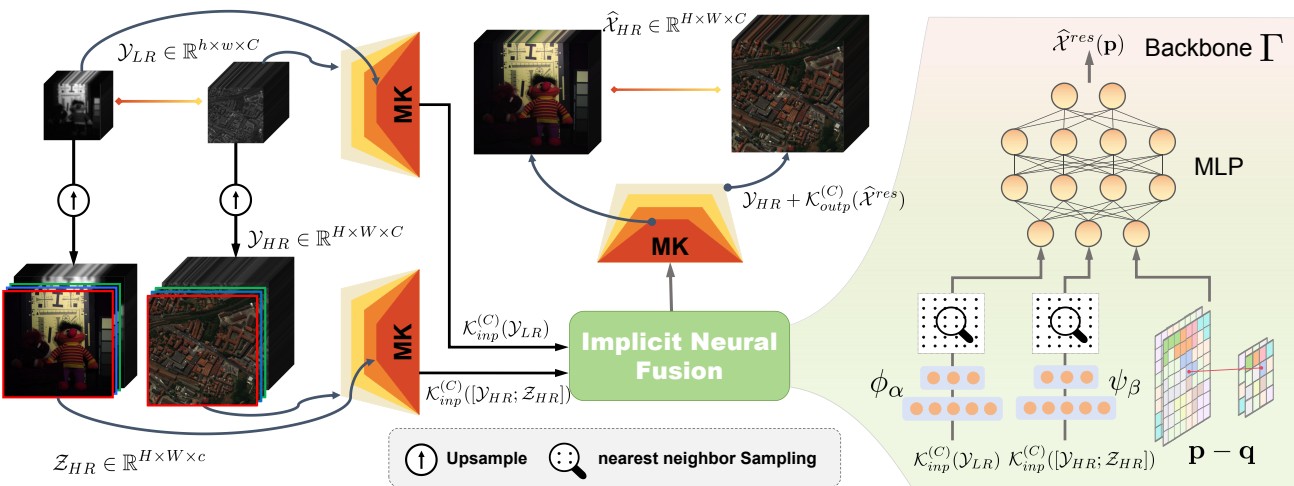

*Figure 2.* The overall architecture of our proposed SSA framework. This end-to-end model realizes the universal mapping function $\mathcal{F}$, which takes an LR-HSI and an HR-MSI from *various sensors* as input and reconstructs a high-fidelity HR-HSI at *arbitrary scales*.

choosing algorithm. The Matryoshka representation is acquired to minimize the empirical risk (that is, a valid representation). Under the classification context and parameterizing the nested $z$ as the classifier weights $\mathbf{W}^{(m)} \in \mathbb{R}^{N,m}$, the empirical risk can be supervised using cross-entropy loss:

$$\min_{\substack{\theta_F, \\ \{\mathbf{W}^{(m)}\}_{m \in \mathcal{M}, \\ m \leq d}}} \frac{1}{N} \sum_{m \in \mathcal{M}} \mathcal{L}\left(\mathbf{W}^{(m)} \cdot F(x, \theta_F)_{1:m}; y\right), \quad (1)$$

where $\mathcal{M}$ is a nested chosen list (*e.g,* $\mathcal{M} := \{m_1, \ldots, m_i, d\}, m_1 \leq m_i \leq d$), $N$ is the total classification number, $(x, y)$ is the data/label pair. In the MRL paper, the sub-representation choosing algorithm is designed as dim-slicing, where $r$ is fixed to 1 and $s = m$.

Unlike MRL, which aims to learn nested representations that can be used at different embedding dimensions (Lin et al., 2024; Wang et al., 2024b), we bring the nested principle into hyperspectral fusion kernels to handle varying spectral channel counts; we term this design Matryoshka Kernels (MK, see § 3.2). This design directly addresses architectural rigidity and helps enable joint training across heterogeneous sensors.

## 3. Method

### 3.1. Problem Formulation and Model Overview

Given a low-resolution HSI (LR-HSI), $\mathcal{Y}_{LR} \in \mathbb{R}^{h \times w \times C}$, and a high-resolution multispectral image (HR-MSI), $\mathcal{Z}_{HR} \in \mathbb{R}^{H \times W \times c}$ ($c \ll C$), the objective of MS/HS fusion is to learn a mapping function $\mathcal{F}_\theta : (\mathcal{Y}_{LR}, \mathcal{Z}_{HR}) \to \widehat{\mathcal{X}}_{HR}$ that reconstructs a high-resolution hyperspectral image (HR-HSI), $\mathcal{X}_{HR} \in \mathbb{R}^{H \times W \times C}$, where $\theta$ represents the learnable parameters of the model.

A key challenge in building a universal fusion model is sensor heterogeneity. For different sensors and optical cameras, the number of spectral bands $C$ and the spatial scaling factor $s$ can vary. This means our training data comes from a collection of datasets $\mathcal{D} = \{D_1, \cdots, D_N\}$, and for any image pair $(\mathcal{Y}_{LR}^{(i)}, \mathcal{Z}_{HR}^{(i)})$ sampled from the $i$-th dataset, it has a corresponding number of spectral bands $C^{(i)}$ and a scaling factor $r^{(i)}$. This requires the universal fusion model should acquire the spectral and spatial agnosticism.

To realize such a universal $\mathcal{F}$, we propose the SSA framework, with its overall architecture depicted in Fig. 2. Our model is designed as an end-to-end, continuous representation-based fusion pipeline. First, the model performs spectral unification given data pair. As shown, any input $\mathcal{Y}_{LR}$, regardless of its original band count $C$, is mapped by a Matryoshka Kernel Layer (MKL) $\mathcal{K}_{inp}^{(C)}(\cdot)$ to a fixed, predefined feature dimension (see §3.2). Concurrently, the HR-MSI $\mathcal{Z}_{HR}$ is concatenated with the spatial upsampled LR-HSI $\mathcal{Y}_{HR} \in \mathbb{R}^{H \times W \times c}$, and this combined tensor is processed by a similar layer.

Subsequently, those two feature maps are fed into the continuous representation fusion backbone (see §3.3). The core of this module is to model an implicit function, parameterized by an MLP, which is responsible for the deep fusion of spectral and spatial information. The continuous feature output by the fusion backbone is then fed to an output MKL $\mathcal{K}_{outp}^{(C)}(\cdot)$ to the pixel space with original band count $C$. Finally, inspired by residual learning in (He et al., 2016), the output $\widehat{\mathcal{X}}^{res}$ is added with the upsampled LR-HSI to produce the final high-fidelity HR-HSI.

## 3.2. Matryoshka Kernels: Spectral Agnosticism

To deal with the drawbacks mentioned § 1, particularly when dealing with inputs of varying channels, we introduce the Matryoshka Kernel Layer (MKL) formulated as $\mathcal{K}_{\star}^{(C)}, \star \in \{inp, outp\}$, which acts as input and output layers. Each MKL contains one Matryoshka-style nested kernel, see Fig. 3. This approach avoids the need for separate layers for each possible channel configuration. For the input MKL, given a hyperspectral cube $\mathbf{X}$ with $C$ bands, it aims to embed the input into the feature $\mathbf{Y}$ with $D$-embedding dimension. For the output MKL, it's vice versa.

Specifically, each MKL internally maintains an MK as the learnable weight. Taking the input MKL weight as example, the MK's weight is as $\mathbf{W}_{\text{nested}} \in \mathbb{R}^{D \times C_{\max} \times k \times k}$, where $D$ is the fixed number of output channels, $C_{\max}$ is the predefined maximum number of input channels the layer can support ($C_{\text{in}} \leq C_{\max}$), and $k \times k$ is the kernel size. A corresponding bias vector $\mathbf{b} \in \mathbb{R}^D$ is also maintained. During the forward pass, a valid kernel is generated from the nested kernel, $\mathbf{W}_{\text{valid}} \in \mathbb{R}^{D \times C_{\text{in}} \times k \times k}$, using a slice-based approach:

$$\mathbf{W}_{\text{valid}} = \mathbf{W}_{\text{nested}}[:, : C_{\text{in}}, :, :], \quad (2)$$

where operation $[:]$ follows Numpy array slice, which selects the first $C_{\text{in}}$ items along the input channel dimension of $\mathbf{W}_{\text{nested}}$ to form the kernel for the current computation. Using the current valid kernel, the final output feature map can be produced by applying a standard 2D convolution:

$$\mathbf{Y} = \texttt{Conv2D}(\mathbf{X}, \mathbf{W}_{\text{valid}}, \mathbf{b}, \text{stride}, \text{padding}). \quad (3)$$

For each element in the $d$-th output channel, the computation is formally expressed as:

$$\mathbf{Y}_{d,i,j} = \mathbf{b}_d + \sum_{c=0}^{C_{\text{in}}-1} \sum_{m=0}^{k-1} \sum_{n=0}^{k-1} \mathbf{X}_{c,i \cdot s+m, j \cdot s+n} \cdot (\mathbf{W}_{\text{valid}})_{d,c,m,n},$$

where $s$ denotes the stride and the indices $d$, $c$, $m$, and $n$ refer to the output channel, input channel, and spatial dimensions of the kernel, respectively.

A similar nested philosophy is applied to the output MKL of the model, enabling it to dynamically generate an output that matches the original band count $C$ of the input sample. During the forward pass, a desired output channel count, $C_{\text{out}}$ ($C_{\text{out}} \leq C_{\max}$), is specified. The valid kernel $\mathbf{W}_{\text{valid}} \in \mathbb{R}^{C_{\text{out}} \times D \times k \times k}$ and bias $\mathbf{b}_{\text{comp}} \in \mathbb{R}^{C_{\text{out}}}$ are generated by slicing the superset parameters along the output channel dimension:

$$\mathbf{W}_{\text{valid}} = \mathbf{W}_{\text{nested}}[: C_{\text{out}}, :, :, :], \quad (4)$$
$$\mathbf{b}_{\text{valid}} = \mathbf{b}_{\text{nested}}[: C_{\text{out}}]. \quad (5)$$

Through this mechanism of a nested kernel and dynamic subset slicing, our SSA model achieves true spectral agnosticism, allowing it to seamlessly process heterogeneous

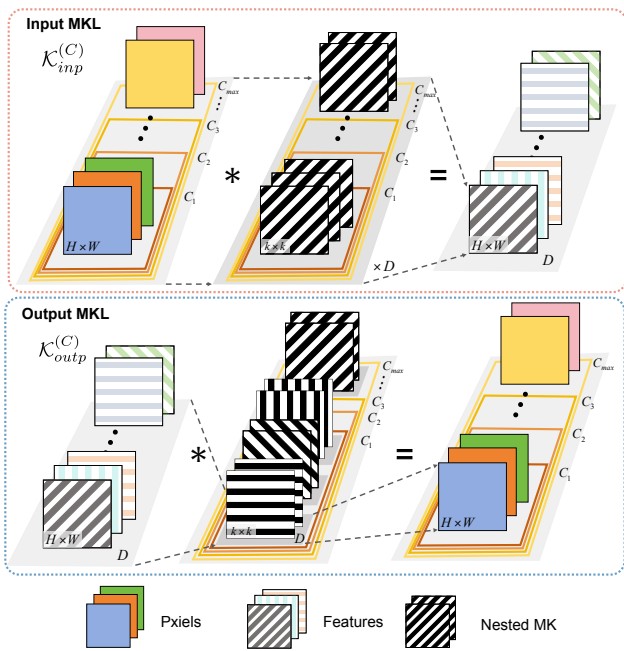

*Figure 3.* Illustration of the MKL's encoding and decoding.

data from any sensor within a universal architecture. The detailed input and output MKL algorithms are shown in App.§ A.

## 3.3. Arbitrary-Scale Image Fusion as Continuous Function

The implementation of our target HR-HSI as a continuous function through the implicit fusion backbone $\Gamma$ is what grants our model true spatial flexibility. It allows the model to be queried on any coordinate grid, enabling the generation of an output at any desired resolution.

As illustrated in Fig. 2 (b), our model first maps the embedded features after MK into a latent space via two parallel encoders $\phi_\alpha, \psi_\beta$. This process can be represented as:

$$\begin{cases} \mathcal{E}_{pe} = \phi_\alpha(\mathcal{K}_{inp}^{(C)}(\mathcal{Y}_{LR})), \\ \mathcal{E}_{pa} = \psi_\beta\left(\mathcal{K}_{inp}^{(C)}([\mathcal{Y}_{HR}; \mathcal{Z}_{HR}])\right), \end{cases}$$

where $[\cdot; \cdot]$ stands for channel concatenation. These latent codes, $\mathcal{E}_{pe}$ and $\mathcal{E}_{pa}$, provide both the spectral and spatial semantic information necessary for the subsequent continuous function modeling.

To predict the spectral vector for any continuous coordinate $\mathbf{p}$ in the target HR-HSI space, our fusion module performs a local, query-based process. For a given pixel, we use its center to represent its coordinate and map the HR coordinates to a normalized square grid $\Omega = [-1, 1]^2$. The integer coordinates $(i, j)$ of a pixel in the $H \times W$ source grid are then mapped to a query coordinate $\mathbf{p}_{(i,j)} \in \Omega \subset \mathbb{R}^2$ using

*Table 1.* The average and standard deviation calculated for all the compared approaches on 10 CAVE examples and 2 the PaviaC dataset examples simulating ×4 (*in-distribution*) and ×8, ×16, ×32 (*out-of-distribution*). Results are highlighted as best and second best .

| | Metric | CAVE Dataset | | | | | | | PaviaC Dataset | | | | | | |
| | | DSPNet | MIMO-SST | DCINN | DCFormer | PSTUN | LRTN | Ours | DSPNet | MIMO-SST | DCINN | DCFormer | PSTUN | LRTN | Ours |
|---|---|---|---|---|---|---|---|---|---|---|---|---|---|---|---|
| In-Dist. ×4 | PSNR(↑) | 40.37±3.25 | 45.35±3.35 | 43.75±3.77 | 45.81±5.56 | 40.57±3.72 | 41.13±3.24 | 45.96±4.69 | 43.89±1.78 | 42.04±0.78 | 44.79±1.99 | 42.61±1.19 | 44.94±1.50 | 45.13±1.01 | 45.92±1.75 |
| | SAM(↓) | 2.72±0.89 | 2.74±0.86 | 2.72±0.81 | 2.70±1.03 | 2.94±0.89 | 2.92±1.38 | 2.55±1.00 | 3.15±0.43 | 3.29±0.45 | 2.59±0.41 | 3.55±0.43 | 2.95±0.46 | 3.18±0.45 | 2.02±0.30 |
| | ERGAS(↓) | 2.51±0.71 | 2.08±0.68 | 2.37±0.95 | 2.01±1.82 | 2.70±0.89 | 2.38±0.82 | 1.72±1.13 | 1.58±0.10 | 1.73±0.02 | 1.34±0.12 | 1.82±0.05 | 1.47±0.07 | 1.48±0.01 | 1.07±0.11 |
| | SSIM(↑) | 0.993±0.005 | 0.994±0.005 | 0.992±0.004 | 0.993±0.004 | 0.992±0.005 | 0.992±0.006 | 0.992±0.005 | 0.988±0.000 | 0.986±0.000 | 0.992±0.000 | 0.983±0.000 | 0.990±0.000 | 0.988±0.000 | 0.996±0.000 |
| Out-of-Dist. ×8 | PSNR(↑) | 36.92±3.47 | 29.73±3.21 | 33.72±3.39 | 41.46±4.98 | 34.69±3.17 | 33.40±3.13 | 38.80±3.79 | 38.03±0.80 | 34.45±0.55 | 32.82±0.58 | 36.57±0.88 | 37.27±0.93 | 33.57±0.40 | 38.59±0.67 |
| | SAM(↓) | 3.82±1.59 | 5.55±2.66 | 4.33±1.94 | 3.65±1.65 | 4.30±1.59 | 4.55±2.03 | 3.70±1.45 | 5.66±0.25 | 6.70±0.14 | 6.09±0.19 | 5.49±0.26 | 5.48±0.20 | 6.67±0.15 | 4.74±0.06 |
| | ERGAS(↓) | 3.91±1.44 | 9.91±4.03 | 6.24±2.62 | 3.08±2.21 | 5.13±1.93 | 6.00±2.03 | 3.46±1.57 | 4.05±0.05 | 4.53±0.07 | 4.53±0.09 | 3.79±0.06 | 3.60±0.06 | 4.51±0.05 | 3.47±0.05 |
| | SSIM(↑) | 0.980±0.015 | 0.904±0.099 | 0.958±0.029 | 0.985±0.014 | 0.964±0.022 | 0.956±0.030 | 0.982±0.013 | 0.947±0.004 | 0.923±0.006 | 0.929±0.005 | 0.948±0.004 | 0.953±0.003 | 0.931±0.006 | 0.960±0.004 |
| ×16 | PSNR(↑) | 30.87±3.46 | 24.76±2.93 | 27.65±3.43 | - | 29.47±3.16 | 27.53±2.97 | 31.84±3.47 | 32.92±0.43 | 31.49±0.20 | 28.74±0.05 | - | 34.28±0.39 | 28.65±0.11 | 35.00±0.02 |
| | SAM(↓) | 6.51±2.83 | 8.36±4.48 | 6.93±3.48 | - | 6.68±2.44 | 6.73±3.08 | 5.93±2.42 | 8.43±0.39 | 8.94±0.13 | 8.72±0.18 | - | 7.48±0.30 | 9.64±0.10 | 6.89±0.08 |
| | ERGAS(↓) | 7.58±2.66 | 16.69±6.56 | 12.19±4.99 | - | 8.85±2.80 | 11.44±4.40 | 7.30±2.88 | 6.79±0.05 | 6.25±0.07 | 6.78±0.17 | - | 5.04±0.04 | 7.39±0.30 | 5.16±0.07 |
| | SSIM(↑) | 0.947±0.032 | 0.834±0.102 | 0.901±0.061 | - | 0.926±0.037 | 0.905±0.057 | 0.953±0.029 | 0.910±0.001 | 0.898±0.003 | 0.880±0.002 | - | 0.932±0.001 | 0.883±0.005 | 0.935±0.002 |
| ×32 | PSNR(↑) | 27.91±2.95 | 21.85±2.56 | 23.96±2.77 | - | 27.55±2.75 | 24.48±2.69 | 28.06±2.64 | 30.51±0.21 | 29.91±0.02 | 26.51±0.17 | - | 32.63±0.46 | 26.53±0.41 | 32.79±0.26 |
| | SAM(↓) | 10.03±3.94 | 12.26±5.90 | 10.34±4.73 | - | 9.43±3.28 | 9.64±3.96 | 8.64±3.17 | 10.26±0.51 | 10.69±0.33 | 10.93±0.47 | - | 9.16±0.66 | 11.89±0.43 | 8.49±0.19 |
| | ERGAS(↓) | 11.12±3.90 | 23.54±8.22 | 18.98±4.18 | - | 12.35±3.30 | 16.80±5.97 | 11.40±6.69 | 8.62±0.57 | 7.39±0.33 | 8.44±0.50 | - | 6.00±0.32 | 9.26±0.85 | 6.24±0.30 |
| | SSIM(↑) | 0.912±0.036 | 0.774±0.099 | 0.843±0.066 | - | 0.896±0.034 | 0.860±0.061 | 0.918±0.032 | 0.893±0.002 | 0.887±0.000 | 0.853±0.003 | - | 0.919±0.004 | 0.856±0.001 | 0.922±0.001 |

the following transformation:

$$\mathbf{p}_{(i,j)} = \left( \frac{2i}{H-1} - 1, \frac{2j}{W-1} - 1 \right). \tag{6}$$

Using the query coordinate $\mathbf{p}$ as a pivot, we sample a precise local spatial feature from the spatial latent code $\mathcal{E}_{pa}$. Concurrently, we identify the nearest neighbor $\mathbf{q}_{(i,j)} \in \Omega$ corresponding to $\mathbf{p}_{(i,j)}$ in the low-resolution spectral latent code $\mathcal{E}_{pe}$ and sample the features of this neighbor.

For the sampled local spectral features, we feed the previous spectral, spatial codes and the relative coordinates $\mathbf{p} - \mathbf{q}$ that describe the fine-grained position into a shared MLP decoder:

$$\widehat{\mathcal{X}}^{res}(\mathbf{p}) = \texttt{MLP}(\mathcal{E}_{pe}(\mathbf{p}), \mathcal{E}_{pa}(\mathbf{q}), \mathbf{p} - \mathbf{q}), \tag{7}$$

where $\mathbf{p}$ is the normalized coordinates of each query pixel in the HR domain. The output of the MLP is then fed into the output MK layer. Leveraging the MK, this layer dynamically maps the MLP feature back to the original spectral band count $C$ of the current sample, generating the final predicted residual components $\mathcal{K}_{outp}^{(C)}(\widehat{\mathcal{X}}^{res}(\mathbf{p}))$. The complete fused image is finally obtained by adding the predicted residual to the bicubic-interpolated base image:

$$\widehat{\mathcal{X}}_{HR}(\mathbf{p}) = \mathcal{Y}_{HR}(\mathbf{p}) + \mathcal{K}_{outp}^{(C)}(\widehat{\mathcal{X}}^{res}(\mathbf{p})). \tag{8}$$

Additionally, to ensure spatial continuity when crossing grid boundaries, we follow the practice of LIIF (Chen et al., 2021b). We identify the four-nearest neighbor grid points $\mathbf{q}_i, i \in \{1, \dots 4\}$ in the low-resolution feature map. During decoding, we predict a candidate spectral residual $\widehat{\mathcal{X}}_i^{res}(\mathbf{p})$ and a scalar weight $w_i$ for each neighbor $q_i$. The final residual vector is then obtained by a softmax weighted fusion of all candidate residuals:

$$\widehat{\mathcal{X}}^{res}(\mathbf{p}) = \sum_{i=1}^{4} \frac{\exp(w_i)}{\sum_{j=1}^{4} \exp(w_j)} \widehat{\mathcal{X}}_i^{res}(\mathbf{p}). \tag{9}$$

True scale agnosticism is achieved through this coordinate-based approach. By modeling the HR-HSI as a continuous function, our architecture avoids fixed upsampling layers, granting it the ability to generate outputs at any resolution.

### 3.4. Cross-Sensor Joint Training Strategy

The conventional "one dataset, one model" paradigm is not only inefficient but also prone to being easily overfitting (Yan et al., 2025), especially given the small scale of HSI datasets. Training a universal model within various hyperspectral data usually faces a challenge that different bands and spatial sizes can not be batched together to enable joint training.

Our training pipeline is designed to handle $N$ heterogeneous datasets, denoted as $\mathcal{D} = \{D_1, \dots, D_N\}$. We first partition all training samples into distinct buckets, where each bucket $D_i$ contains data from a single source dataset, thus sharing identical spatial dimensions and number of spectral bands. At each training step, we construct a mini-batch by a two-stage sampling process. First, a bucket $D_i$ is selected from $\mathcal{D}$ following a uniform random distribution. Then, a batch of samples is drawn exclusively from the chosen bucket. This strategy ensures that all samples within a single mini-batch are structurally homogeneous, while samples across different mini-batches can be heterogeneous. It is precisely this batch-to-batch heterogeneity that our model is designed for. Crucially, all samples are processed by the same set of model parameters $\theta$, which are optimized via a unified hybrid reconstruction loss function. Following Eq. (1), the reconstruction loss can be written in the MRL style:

$$\min_{\substack{\theta, C \in \text{Dim}(D_i), \\ (\mathcal{Y}_{LR}, \mathcal{Z}_{HR}) \in D_i}} \frac{1}{N} \sum_{i=1}^{N} \mathcal{L}_{total} \left[ \mathcal{F}_{\theta}(\mathcal{Y}_{LR}, \mathcal{Z}_{HR}); \mathcal{X}_{HR} \right],$$

where $\mathcal{F}_{\theta}$ is the model (*i.e.*, $\mathcal{F}_{\theta} := \mathcal{K}_{outp}^{(C)} \circ \Gamma \circ \mathcal{K}_{inp}^{(C)}$ and $\circ$ means the network composition), $\text{Dim}(\cdot)$ denotes the bands numbers in the data bucket. Our total loss, $\mathcal{L}_{total}$, is a weighted sum of the L1 loss and the Structural Similarity

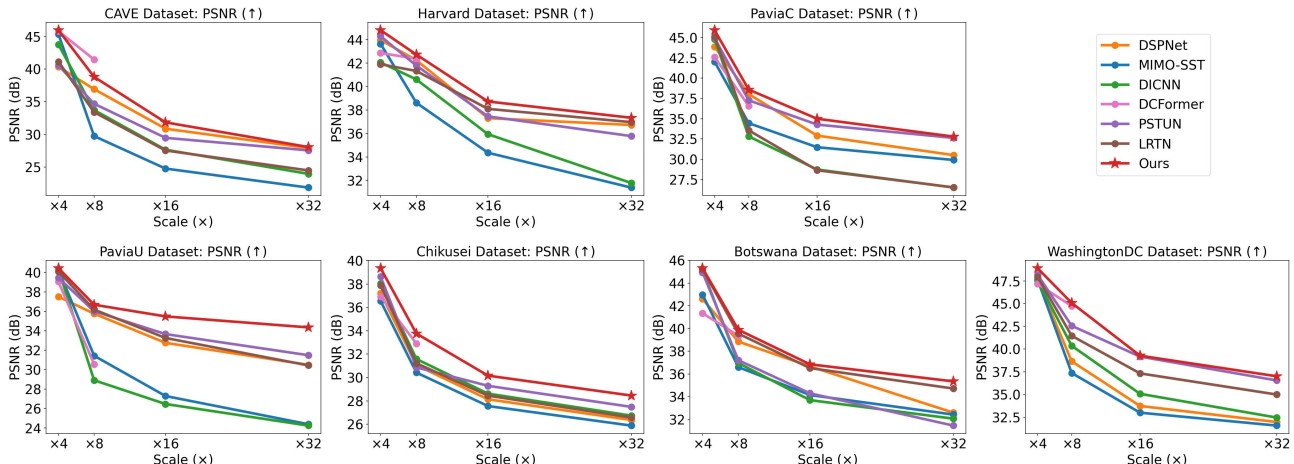

*Figure 4.* Quantitative comparison of PSNR (dB) across multiple scaling factors on all seven datasets.

(SSIM) loss, defined as:

$$\mathcal{L}_{total} = \mathbb{E}_{(\mathcal{Y}_{LR}, \mathcal{Z}_{HR}, \mathcal{X}_{HR}) \sim D_i} [\mathcal{L}_1 + \lambda \mathcal{L}_{SSIM}],$$
$$\mathcal{L}_1 = \|\mathcal{F}_\theta(\mathcal{Y}_{LR}, \mathcal{Z}_{HR}) - \mathcal{X}_{HR}\|_1, \quad (10)$$
$$\mathcal{L}_{SSIM} = 1 - \mathrm{SSIM}(\mathcal{F}_\theta(\mathcal{Y}_{LR}, \mathcal{Z}_{HR}), \mathcal{X}_{HR}),$$

where $\lambda$ is a weighting coefficient, $\|\cdot\|$ is the $\ell_1$ norm, and $\mathrm{SSIM}(\cdot, \cdot)$ is the structural similarity index measure.

Through this joint training strategy, SSA is encouraged to learn a shared fusion mapping rather than memorizing the specifics of a particular dataset. This approach supports transfer to sensors not seen during joint training (see § 4.4.2).

## 4. Experiments

### 4.1. Datasets

To evaluate our model's performance and generalization, we use a diverse suite of seven public hyperspectral datasets, covering *indoor, outdoor, and remote sensing* scenarios. Dataset information is summarized in Tab. B.1.

Our setup deliberately embraces heterogeneity, with spectral bands ranging from 31 (CAVE) to 191 (Washington DC), and substantially differing spatial resolutions and spectral ranges from sensors like ROSIS and Hyperion. This diversity provides the perfect testbed for validating the spectral agnosticism of our SSA framework. We followed standard train/test splits for each dataset, using appropriate patch sizes as detailed in the table.

### 4.2. Implementation Details

Our SSA model and all experiments were implemented using the PyTorch framework (Paszke et al., 2019). All training and inference were conducted on a single NVIDIA RTX

4090 GPU. For the training process, we use the AdamW optimizer (Loshchilov & Hutter, 2017) and trained the model for 2000 epochs with a batch size of 4. The learning rate was managed by a cosine annealing learning rate scheduler, starting from 2e-4 and annealing to 1e-5. To enable the spectral agnosticism of MK, the crucial hyperparameter for the maximum channel, $C_{\max}$, was set to 194. Both encoders are based on the EDSR architecture (Lim et al., 2017).

### 4.3. Main Results

To highlight the fundamental difference between our proposed model and existing methods, for each dataset (at a $4\times$ ratio), we trained a dedicated model independently for all SOTA methods to be compared, and evaluated them on the test sets with ratios of $4\times$, $8\times$, $16\times$, and $32\times$. For our method, we trained only a single, unified model, with the training set composed of a mixture of training samples from the seven datasets. The evaluation on the test set was conducted in the same manner as before. Additionally, we adopted four widely used quality metrics, namely Peak Signal-to-Noise Ratio (PSNR), Spectral Angle Mapper (SAM), Erreur Relative Globale Adimensionnelle de Synthèse (ERGAS), and the Structural Similarity (SSIM). The calculation formulas for the above indicators can be found in the App.§ B.3.

#### 4.3.1. QUANTITATIVE COMPARISON

Tab. 1 presents the quantitative comparison of all methods on the CAVE and PaviaC datasets, under both in-distribution (the $\times 4$ training scale) and out-of-distribution ($\times 8$, $\times 16$, and $\times 32$ scales) settings. The data in the table indicates that our method achieves SOTA performance across all in-distribution metrics. In the out-of-distribution evaluation, our model secures the best or second-best results on nearly all out-of-distribution metrics. Although methods like DC-

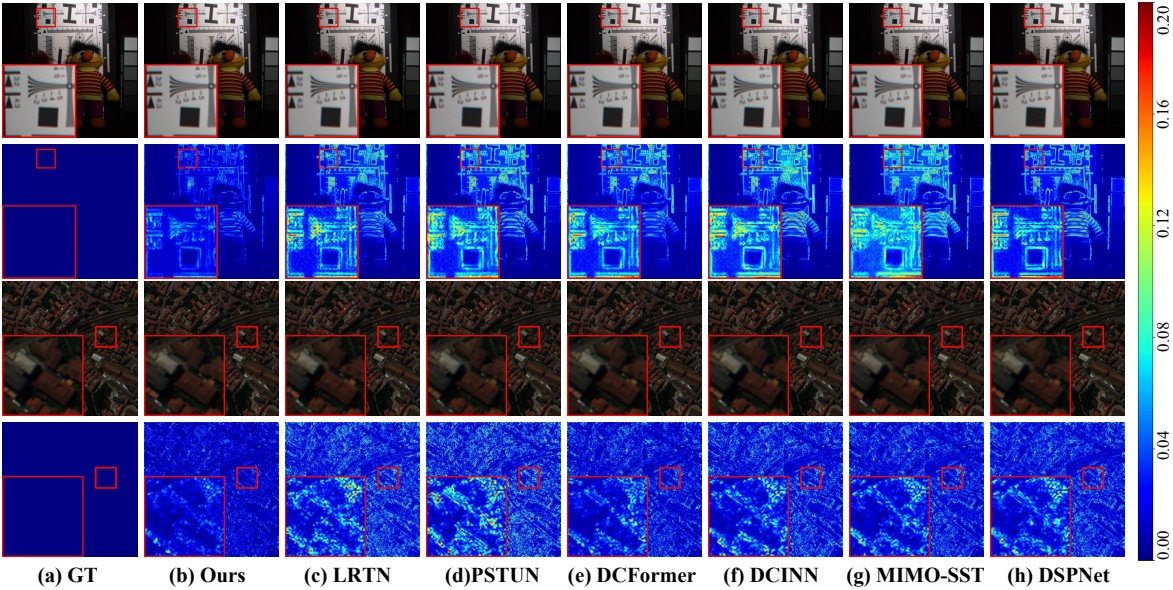

(a) GT  (b) Ours  (c) LRTN  (d)PSTUN  (e) DCFormer  (f) DCINN  (g) MIMO-SST  (h) DSPNet

*Figure 5.* The upper and lower parts respectively showcase the fused images and error maps from the CAVE and PaviaC datasets. Red rectangles depict some close-up shots.

*Table 2.* Ablation study of MK on the PaviaC dataset (See §4.4.1).

| | Model | PSNR(↑) | SAM(↓) | ERGAS(↓) | SSIM(↑) |
|---|---|---|---|---|---|
| ×4 | w/o MK | 45.85 ± 1.80 | **2.01 ± 0.29** | **1.06 ± 0.11** | 0.995 ± 0.001 |
| | **Ours** | **45.92 ± 1.75** | 2.02 ± 0.30 | 1.07 ± 0.11 | **0.996 ± 0.000** |
| ×8 | w/o MK | **38.62 ± 0.65** | 4.75 ± 0.07 | 3.49 ± 0.06 | 0.959 ± 0.004 |
| | **Ours** | 38.59 ± 0.67 | **4.74 ± 0.06** | **3.47 ± 0.05** | **0.960 ± 0.004** |
| ×16 | w/o MK | 34.95 ± 0.03 | 6.92 ± 0.08 | 5.18 ± 0.07 | **0.936 ± 0.002** |
| | **Ours** | **35.00 ± 0.02** | **6.89 ± 0.08** | 5.16 ± 0.07 | 0.935 ± 0.002 |
| ×32 | w/o MK | **32.81 ± 0.25** | 8.51 ± 0.20 | 6.25 ± 0.29 | 0.921 ± 0.001 |
| | **Ours** | 32.79 ± 0.26 | **8.49 ± 0.19** | **6.24 ± 0.30** | **0.922 ± 0.001** |

*Table 3.* Controlled comparison under the same pooled 7-dataset joint-training regime. Results are macro-averaged over the 7 datasets at each scale; Overall is averaged over all dataset-scale settings.

| Model | Interface | ×4 | ×8 | ×16 | ×32 | Overall |
|---|---|---|---|---|---|---|
| Joint w/o MK | Padding+mask | 43.74 | 39.08 | 35.02 | 32.72 | 37.64 |
| Joint w/ adapter | 1 × 1 projection | 44.03 | 39.31 | 35.29 | 33.05 | 37.92 |
| **Ours** | **MK** | **44.58** | **39.91** | **36.09** | **33.90** | **38.62** |

Former (Wu et al., 2025), DSPNet (Sun et al., 2023), and PSTUN (Wang et al., 2025a) also achieve strong second-best results in some specific cases, they suffer from the inability to handle all scales and exhibit inconsistent performance. The quantitative comparison results for the other five datasets can be found in App.§ B.7. To provide a comprehensive view of the performance across all datasets, we plot the PSNR metrics of all methods in a line chart (see Fig. 4). It can be observed that our model does not compromise on reconstruction quality; its performance can even surpass that of specialized models that were individually trained for each specific dataset. Our method provides the best in-distribution accuracy and demonstrates robust generalization to unseen scales.

### 4.3.2. QUALITATIVE COMPARISON

To illustrate the advantages of our method, we provide a visual comparison in Fig. 5 with SOTA approaches, including close-ups and error maps to highlight specific details. Visually, the results generated by our SSA model are markedly superior and most faithful to the GT. Visually, our SSA model yields results most faithful to the ground truth. The close-up views reveal that our method effectively recovers fine details, such as the texture of the stuffed toy and crisp lines on the chart, while competitors often suffer from blurriness and artifacts. Similarly, it reconstructs clearer architectural structures in the remote sensing scene. This superiority is corroborated by the error maps: while ours remain predominantly dark blue (indicating minimal residuals), SOTA methods display significant high-energy areas (yellow/red) along object boundaries, demonstrating the robust accuracy of our unified approach.

### 4.4. Ablation Studies

#### 4.4.1. THE NECESSITY OF MATRYOSHKA KERNELS

To validate the role of MK, we evaluate a w/o MK variant by replacing our MK layers with standard fixed-channel convolutions and training it independently on each dataset.

Tab. 2 compares this individually trained variant against our single, jointly trained model. The results show that the

*Table 4.* **Generalization to Unseen Bands and Fractional Scales.** Top: generalization on unseen datasets at ×4 scale, including Houston and Loukia. Bottom: generalization on unseen scales on CAVE (trained at ×4 ratio). Results of DCFormer/PSTUN/LRTN are obtained by bicubic resizing based on ×4 ratio.

| Model | PSNR(↑) | SAM(↓) | ERGAS(↓) | SSIM(↑) |
|---|---|---|---|---|
| *Unseen Dataset — Houston (48 bands, ×4 scale)* | | | | |
| DCFormer | 38.55 | 3.21 | 4.09 | 0.972 |
| PSTUN | 38.52 | 3.15 | 3.98 | 0.975 |
| LRTN | 38.65 | 3.12 | 3.91 | **0.980** |
| **Ours** | **38.71** | **3.09** | **3.85** | 0.978 |
| *Unseen Dataset — Loukia (176 bands, ×4 scale)* | | | | |
| DCFormer | 36.25 | **3.88** | 3.65 | 0.958 |
| PSTUN | 35.80 | 4.12 | 3.82 | 0.952 |
| LRTN | 35.92 | 4.05 | 3.78 | 0.955 |
| **Ours** | **36.42** | 3.95 | **3.55** | **0.962** |
| *Fractional Scales — CAVE (trained only at ×4) ×3.2 scale* | | | | |
| Bicubic | 38.50 | 5.12 | 4.85 | 0.945 |
| DCFormer | 44.65 | **2.45** | 1.95 | 0.990 |
| PSTUN | 43.90 | 2.62 | 2.25 | 0.985 |
| LRTN | 44.35 | 2.58 | 2.08 | 0.988 |
| **Ours (direct)** | **45.02** | 2.48 | **1.81** | **0.993** |
| *Fractional Scales — CAVE (trained only at ×4) ×5.7 scale* | | | | |
| Bicubic | 32.20 | 6.85 | 6.10 | 0.895 |
| DCFormer | 42.95 | 3.02 | 2.45 | 0.984 |
| PSTUN | 41.25 | 3.25 | 2.72 | 0.980 |
| LRTN | 42.80 | **2.88** | 2.55 | 0.982 |
| **Ours (direct)** | **43.49** | 2.91 | **2.17** | **0.991** |

performance between the two models is highly comparable across all scales. While the 'w/o MK' variant occasionally performs marginally better, our unified model with MK often matches or exceeds its performance. This demonstrates that our model achieves performance on par with specialized models.

To isolate the role of MK under joint training, we include controlled baselines under the exact same pooled 7-dataset regime, as shown in Tab. 3. We keep the backbone, INR decoder, optimizer, schedule, and bucket-based sampling unchanged, and replace only the input/output MK layers with: (i) fixed-channel convolutions at $C_{max}$ with zero-padding and a valid-loss mask, and (ii) a learned input/output $1 \times 1$ spectral adapter. The mask is used only to ignore padded output channels in the loss and is not fed to the network. Both baselines are consistently worse than SSA in average PSNR: the zero-padding baseline drops by about 1.0 dB, and the learned $1 \times 1$ adapter still trails by about 0.7 dB. This provides direct evidence that pooled-data training alone is not sufficient, and that even a stronger learned fixed-band adapter remains weaker than MK under the same joint-training protocol. MK should therefore be understood primarily as the mechanism enabling effective universal joint training, rather than as a claim of unconditional superiority over separately trained fixed-band models.

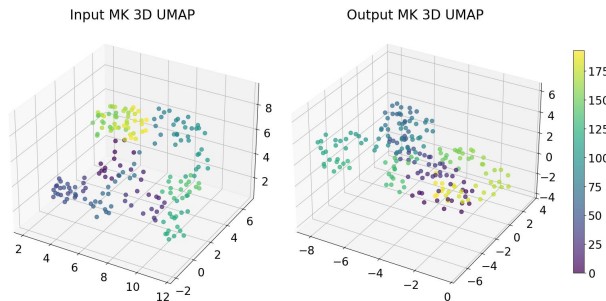

*Figure 6.* UMAP (McInnes et al., 2018) visualization of the learned MK weights, which form a smooth manifold that reveals the model's encoding of spectral continuity, key to its generalization.

### 4.4.2. UNSEEN BANDS AND FRACTIONAL SCALES

We further evaluate the generalization of our pretrained model under two challenging settings: unseen datasets with different spectral bands and unseen fractional scales.

**Unseen datasets (unseen bands).** We first test on two datasets whose spectral configurations are not observed during training: Houston with 48 bands and Loukia with 176 bands, both evaluated at the ×4 scale. As shown in Tab. 4, our method, fine-tuned for only 500 iterations, consistently achieves top-tier performance across metrics. On Houston, our model obtains the best PSNR/SAM/ERGAS (38.71 dB / 3.09 / 3.85) and remains competitive in SSIM (0.978), demonstrating robust generalization to a different band setup. On Loukia, our method again delivers the best reconstruction quality, achieving the highest PSNR (36.42 dB), the lowest ERGAS (3.55), and the highest SSIM (0.962), while remaining competitive in SAM. These results indicate that our learned spectral representation transfers well across datasets with varying band numbers and distributions.

**Unseen fractional scales.** We further evaluate scale generalization by testing on CAVE at fractional scales ×3.2 and ×5.7, while training is performed only at the integer scale ×4. For methods without native continuous scaling, we follow a strong baseline protocol: generate the ×4 output and then resize to the target fractional scale via Bicubic interpolation (×4 →Bicubic). Tab. 4 shows that our method directly predicts fractional scales and achieves the best PSNR at both scales (45.02 dB at ×3.2 and 43.49 dB at ×5.7), together with the lowest ERGAS (1.81 and 2.17) and the highest SSIM (0.993 and 0.991). Overall, these results confirm that our continuous representation provides genuine scale-agnostic capability, generalizing effectively beyond the discrete scale seen during training and outperforming strong ×4→Bicubic baselines.

### 4.4.3. ANALYSIS OF LEARNED KERNEL STRUCTURES

To investigate the internal working mechanisms of our MKs, we visualized the weight structures of both the input and out-

put kernels using UMAP (McInnes et al., 2018), as shown in Fig. 6. Each point in the visualizations corresponds to a spectral channel, colored by its index. As can be seen, the projected points form smooth and continuous manifolds rather than a randomly scattered cloud. This indicates that the learning process of the kernels has formed a meaningful, layered manifold structure, where the network actively captures the high correlation between adjacent spectral bands. This learned internal structure is key to our model's ability to generalize effectively across different sensors.

## 4.5. Efficiency Analysis

We conducted efficiency analyses on all the comparison models and ablation experiments on our model scaling, which are detailed in App.§ B.4 and App.§ B.5.

## 5. Conclusion

In this work, we addressed the prevalent rigidity of MS/HS fusion models by proposing SSA, a unified framework that combines Matryoshka Kernels (MK) with an Implicit Neural Representation (INR) backbone to mitigate architectural rigidity, data scarcity, and fixed spatial scaling. Our experiments demonstrate that a single jointly trained SSA model can achieve state-of-the-art performance, generalize to unseen spatial scales, transfer to unseen sensors with few-step adaptation, and learn meaningful internal representations. While INR-based decoding introduces increasing computational cost at high output resolutions, SSA provides a practical step toward more flexible and broadly applicable hyperspectral fusion models.

## Acknowledgements

This work was supported by the Fundamental and Interdisciplinary Disciplines Breakthrough Plan of the Ministry of Education of China (JYB2025XDXM102), the National Natural Science Foundation of China (Grants 62576080 and 62220106008), the Guangdong Introducing Innovative and Entrepreneurial Teams (Grant 2023ZT10×044), and the Shenzhen Science and Technology Research Fund (Grant JCYJ20220818103001002).

## Impact Statement

This work aims to advance machine learning methods for multispectral and hyperspectral image fusion. Potential beneficial applications include environmental monitoring, precision agriculture, scientific imaging, and medical or industrial inspection, where improved spatial and spectral reconstruction can support downstream analysis. The same capability may also lower deployment barriers in sensitive settings such as surveillance, military target analysis, or other forms of remote sensing with direct consequences for individuals or communities.

The scope of the proposed agnosticism is limited. Spectral-band agnosticism means that one model can process inputs with different ordered spectral band counts without architectural redesign, as long as $C \leq C_{\max}$; scale agnosticism means that the same model can be queried at arbitrary, including non-integer, scales. The model is not permutation-invariant and relies on physically meaningful wavelength ordering. It was not designed or evaluated for extreme spectral dimensionalities without increasing $C_{\max}$ and retraining, and performance may degrade under stronger sensor-response mismatch, real-world noise, or conditions substantially outside the current benchmark suite. Cross-sensor transfer should therefore be validated for the target sensor and task before deployment, especially in high-stakes applications. In addition, INR-based decoding becomes more expensive as the number of queried HR pixels grows, which may limit use in latency- or memory-constrained settings.

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

# A. Matryoshka Kernel Layer Algorithms

The pseudocode for our proposed Matryoshka Kernel Layer (MKL), as detailed in §3.2, is presented in Algo. 1 and Algo. 2. In our SSA model, both the Matryoshka Kernel Input and Output Layers utilize a $3 \times 3$ convolution kernel. The stride, dilation, and groups are all set to 1. Consequently, the spatial dimensions of the input and output feature maps remain unchanged across the MKL operations.

---

**Algorithm 1** Matryoshka Kernel Input Layer Forward Pass

---

**Require:**
**input** $\mathbf{X} \in \mathbb{R}^{N \times C_{\text{in}} \times H \times W}$
**input** Nested convolution weight $\mathbf{W}_{\text{nested}} \in \mathbb{R}^{D \times C_{\text{max}} \times k \times k}$
**input** Convolution bias $\mathbf{b} \in \mathbb{R}^{D}$ (optional)
**input** Convolution parameters: stride $\mathbf{s}$, padding $\mathbf{p}$, dilation $\mathbf{d}$, groups $\mathbf{g}$
**Ensure:**
**output** $\mathbf{Y} \in \mathbb{R}^{N \times D \times H' \times W'}$
1: **function** Forward($\mathbf{X}, \mathbf{W}_{\text{nested}}, \mathbf{b}$)
2:     **if** $C_{\text{in}} = C_{\text{max}}$ **then**
3:         $\mathbf{Y} \leftarrow \texttt{Conv2d}(\mathbf{X}, \mathbf{W}_{\text{nested}}, \mathbf{b}, \mathbf{s}, \mathbf{p}, \mathbf{d}, \mathbf{g})$ {Use full weights}
4:     **else**
5:         $\mathbf{W}_{\text{valid}} \leftarrow \mathbf{W}_{\text{nested}}[:, : C_{\text{in}}, :, :]$ {Slice weights for input channels $C_{\text{in}}$}
6:         $\mathbf{Y} \leftarrow \texttt{Conv2d}(\mathbf{X}, \mathbf{W}_{\text{valid}}, \mathbf{b}, \mathbf{s}, \mathbf{p}, \mathbf{d}, \mathbf{g})$
7:     **end if**
8:     **return** $\mathbf{Y}$
9: **end function**

---

---

**Algorithm 2** Matryoshka Kernel Output Layer Forward Pass

---

**Require:**
**input** $\mathbf{Y} \in \mathbb{R}^{N \times D \times H' \times W'}$
**input** Target output channels $C_{\text{out}}$
**input** Pre-defined convolution weight $\mathbf{W}_{\text{nested}} \in \mathbb{R}^{C_{\text{max}} \times D \times k \times k}$
**input** Pre-defined convolution bias $\mathbf{b}_{\text{nested}} \in \mathbb{R}^{C_{\text{max}}}$ (optional)
**input** Convolution parameters: stride $\mathbf{s}$, padding $\mathbf{p}$, dilation $\mathbf{d}$, groups $\mathbf{g}$
**Ensure:**
**output** $\mathbf{X} \in \mathbb{R}^{N \times C_{\text{out}} \times H \times W}$
1: **function** Forward($\mathbf{Y}, C_{\text{out}}, \mathbf{W}_{\text{nested}}, \mathbf{b}_{\text{nested}}$)
2:     **if** $C_{\text{out}} = C_{\text{max}}$ **then**
3:         $\mathbf{X} \leftarrow \texttt{Conv2d}(\mathbf{Y}, \mathbf{W}_{\text{nested}}, \mathbf{b}_{\text{nested}}, \mathbf{s}, \mathbf{p}, \mathbf{d}, \mathbf{g})$
4:     **else**
5:         $\mathbf{W}_{\text{valid}} \leftarrow \mathbf{W}_{\text{nested}}[: C_{\text{out}}, :, :, :]$ {Slice weights for output channels $C_{\text{out}}$}
6:         $\mathbf{b}_{\text{valid}} \leftarrow$ None
7:         **if** $\mathbf{b}_{\text{nested}} \neq$ None **then**
8:             $\mathbf{b}_{\text{valid}} \leftarrow \mathbf{b}_{\text{nested}}[: C_{\text{out}}]$ {Slice the bias}
9:         **end if**
10:       $\mathbf{X} \leftarrow \texttt{Conv2d}(\mathbf{Y}, \mathbf{W}_{\text{valid}}, \mathbf{b}_{\text{valid}}, \mathbf{s}, \mathbf{p}, \mathbf{d}, \mathbf{g})$
11:     **end if**
12:     **return** $\mathbf{X}$
13: **end function**

---

# B. Additional Experiments Details

### B.1. Information of Datasets

In this section, we provide the information of used dataset, detailed in Tab. B.1.

*Table B.1.* Detailed meta information and train/test splits of the experimental datasets.

| Property | CAVE | Harvard | PaviaC | PaviaU | Chikusei | Botswana | WashingtonDC |
|---|---|---|---|---|---|---|---|
| Bands | 31 | 31 | 102 | 103 | 128 | 145 | 191 |
| Pixel Resolution (m) | 0.001 | 0.001 | 1.3 | 1.3 | 2.5 | 30 | 2.5 |
| Spectral Range (nm) | 400–700 | 420–720 | 430–860 | 430–860 | 363–1018 | 400–2500 | 400–2500 |
| Sensors | Cooled CCD | Nuance FX | ROSIS | ROSIS | Headwall Hyperspec-VNIR-C | Hyperion | Hydice |
| Size | 512×512 | 1040×1392 | 1096×715 | 610×340 | 2517×2335 | 1476×256 | 1208×307 |
| Train/Test Patch Size | 128×128/ 512×512 | 128×128/ 1024×1024 | 128×128/ 256×256 | 128×128/ 512×512 | 128×128/ 1024×1024 | 128×128/ 256×256 | 128×128/ 256×256 |
| Train/Test Splits | 3718/10 | 5360/10 | 2257/2 | 434/2 | 5175/4 | 765/5 | 876/5 |

## B.2. Data Simulation

To comprehensively evaluate the effectiveness of our proposed method, we conducted experiments on seven public hyperspectral datasets, including two indoor/outdoor datasets, CAVE[1] and Harvard[2], and five remote sensing datasets: PaviaC[3], PaviaU[4], Chikusei[5], Botswana[6], and WashingtonDC[7]. All data simulation procedures strictly comply with Wald's Protocol (Wald, 2000). Detailed parameters for each dataset are summarized in Tab. B.1. The corresponding test sets were constructed from non-overlapping image blocks, specifically four 512×512 blocks for Chikusei and 256×256 blocks for the other four remote sensing datasets. For all training sets, we generated 128×128 image patches with a fixed stride. The low-resolution hyperspectral images (LR-HSIs) were synthesized by downsampling the original HSI by a factor of $s$ using an anti-aliasing filter, while the corresponding high-resolution multispectral images (HR-MSIs) were simulated using the spectral response functions of sensors such as ROSIS and Landsat-8.

## B.3. Quality Indexes

To quantitatively assess the performance of hyperspectral image reconstruction, we adopt four widely used quality indexes: PSNR, SAM, SSIM, and ERGAS. These metrics jointly evaluate the spatial fidelity, spectral accuracy, structural similarity, and global reconstruction consistency.

PEAK SIGNAL-TO-NOISE RATIO (PSNR)

PSNR is a spatial metric that measures the ratio between the maximum possible power of a signal and the power of noise corrupting its representation. It provides an effective way to evaluate the spatial quality of reconstructed HSIs. For a reconstructed image $\widehat{\mathcal{X}}_{HR}$ and ground truth $\mathcal{X}_{HR}$, PSNR is defined as:

$$PSNR(\widehat{\mathcal{X}}_{HR}, \mathcal{X}_{HR}) = 20 \log \left( \frac{255}{RMSE(\widehat{\mathcal{X}}_{HR}, \mathcal{X}_{HR})} \right),$$

where RMSE denotes the root mean squared error:

$$RMSE(\widehat{\mathcal{X}}_{HR}, \mathcal{X}_{HR}) = \frac{\|\widehat{\mathcal{X}}_{HR} - \mathcal{X}_{HR}\|_F}{\sqrt{b \cdot m_l}}, \tag{11}$$

in which $b$ is the number of spectral bands, and $m_l$ is the spatial pixel count. A higher PSNR value indicates a more accurate reconstruction.

---

[1] https://cave.cs.columbia.edu/repository/Multispectral
[2] http://vision.seas.harvard.edu/hyperspec/index.html
[3] https://www.ehu.eus/ccwintco/index.php/Hyperspectral_Remote_Sensing_Scenes#Pavia_Centre_scene
[4] https://www.ehu.eus/ccwintco/index.php/Hyperspectral_Remote_Sensing_Scenes#Pavia_University_scene
[5] https://www.sal.t.u-tokyo.ac.jp/hyperdata/
[6] https://www.ehu.eus/ccwintco/index.php/Hyperspectral_Remote_Sensing_Scenes#Botswana
[7] https://engineering.purdue.edu/~biehl/MultiSpec/hyperspectral.html

SPECTRAL ANGLE MAPPER (SAM)

SAM is a spectral metric that computes the angle between two spectral vectors. It is commonly used to assess spectral preservation, ensuring that the spectral shape is well retained during reconstruction. SAM is formulated as:

$$SAM(\widehat{\mathcal{X}}_{HR}, \mathcal{X}_{HR}) = \frac{1}{b} \sum_{j=1}^{b} \arccos\left( \frac{\langle \widehat{x}_j, x_j \rangle}{\|\widehat{x}_j\|\|x_j\|} \right),$$

where $\widehat{x}_j$ and $x_j$ denote the reconstructed and GT spectral vectors of the $j$-th band.

STRUCTURAL SIMILARITY INDEX MEASURE (SSIM)

SSIM evaluates the structural similarity between the reconstructed image and the ground truth. It jointly considers luminance, contrast, and structural information, and is widely adopted for perceptual image quality assessment. For each band, SSIM is defined as:

$$SSIM(\widehat{x}, x) = \frac{(2\mu_{\widehat{x}}\mu_x + C_1)(2\sigma_{\widehat{x}x} + C_2)}{(\mu_{\widehat{x}}^2 + \mu_x^2 + C_1)(\sigma_{\widehat{x}}^2 + \sigma_x^2 + C_2)},$$

where $\mu$ and $\sigma$ denote the mean and variance, $\sigma_{\widehat{x}x}$ is the covariance, and $C_1$, $C_2$ are constants preventing instability. A higher SSIM value (maximum 1) indicates better structural similarity.

ERREUR RELATIVE GLOBALE ADIMENSIONNELLE DE SYNTHÈSE (ERGAS)

ERGAS is a global index that measures the relative reconstruction error across all spectral bands. Lower ERGAS values indicate better overall reconstruction performance. ERGAS is defined as:

$$ERGAS(\widehat{\mathcal{X}}_{HR}, \mathcal{X}_{HR}) = 100 \cdot \frac{1}{s} \sqrt{\frac{1}{b} \sum_{j=1}^{b} \left( \frac{RMSE_j}{\mu_j} \right)^2},$$

where $s$ is the scale factor, $RMSE_j$ is the per-band RMSE (eq. 11) between $\widehat{\mathcal{X}}_{HR}$ and $\mathcal{X}_{HR}$, and $\mu_j$ is the mean of the GT band.

### B.4. Efficiency Analysis

Tab. B.2 summarizes the model complexity and practical inference efficiency of representative fusion methods. For data-driven deep learning–based hyperspectral fusion, increasing model capacity is often natural, and sometimes necessary, to effectively leverage large-scale heterogeneous training data and to improve generalization. Accordingly, we intentionally scale up the network to enhance its fitting ability. Even so, our model still contains fewer parameters than PSTUN (Wang et al., 2025a) while achieving higher reconstruction accuracy.

From the complexity perspective, Tab. B.2 together with Tab. 1 suggests a consistent trend: methods with larger computational budgets generally deliver stronger reconstruction quality both in-distribution and out-of-distribution, whereas very lightweight models (e.g., DCFormer (Wu et al., 2025)) tend to suffer from noticeable quality degradation due to limited capacity.

Importantly, the increased capacity does not translate into prohibitive runtime. Although our method involves higher theoretical FLOPs, it remains efficient in practice on modern GPUs: at $\times 4$ and $512 \times 512$ input, our model runs in 83.4 ms per image (3.1 MPix/s) with a peak memory footprint of 2.46 GB. This runtime is comparable to (or faster than) several strong baselines with similar or smaller capacity (e.g., PSTUN and LRTN), while being orders of magnitude faster than query-based baselines such as DCINN (Wang et al., 2024a). Overall, these results indicate that scaling the model is an effective way to improve accuracy and robustness, without sacrificing practical inference efficiency.

We further include a patch-level scaling study (LR input size = $32 \times 32$, HR output size = round($32 \times$ scale), batch size = 1) to isolate how runtime and memory change with output resolution, as shown in Tab. B.3. No coordinate subsampling is used in any reported efficiency result: inference is performed at full target resolution. At high resolutions, tiling may be employed solely for memory control, but each tile is still decoded with dense coordinate prediction rather than subsampling. As expected for query-based decoding, runtime increases substantially with scale while throughput gradually decreases. Peak VRAM is not strictly monotonic; for example, the lower peak at $\times 16$ than at $\times 8$ is due to a switch to a tiled execution path to keep memory manageable.

*Table B.2.* Model complexity and inference efficiency on RTX 4090. Time/VRAM/MPix/s are measured at $\times 4$ with batch size 1 and input size $512 \times 512$ ($C = 191$). We report the mean runtime over 50 iterations after 20 warm-up runs using each method's validation forward entrypoint. AMP (FP16) is enabled when supported; otherwise FP32 is used. Peak VRAM is the maximum `cuda.max_memory_allocated` during inference. Throughput is computed as $(HW/10^6)/$(time in seconds).

| Method | Params (M)↓ | FLOPs (G)↓ | Time (ms)↓ | Peak VRAM (GB)↓ | Throughput MPix/s↑ |
|---|---|---|---|---|---|
| DSPNet (Sun et al., 2023) | 6.056 | 27.279 | 38.7 | 1.52 | 6.8 |
| MIMO-SST (Fang et al., 2024) | 4.983 | 6.164 | 14.4 | 1.22 | 18.2 |
| DCINN (Wang et al., 2024a) | 2.417 | 51.531 | 860.9 | 2.25 | 0.3 |
| DCFormer (Wu et al., 2025) | 0.082 | 0.579 | 40.2 | 1.00 | 6.5 |
| PSTUN (Wang et al., 2025a) | 29.948 | 102.400 | 102.4 | 1.58 | 2.6 |
| LRTN (Liu et al., 2025) | 3.691 | 8.396 | 153.7 | 3.32 | 1.7 |
| Ours | 18.654 | 206.848 | 83.4 | 2.46 | 3.1 |

*Table B.3.* Patch-level scaling study of arbitrary-scale inference.

| Scale | Time (ms)↓ | Peak VRAM (GB)↓ | Throughput MPix/s↑ |
|---|---|---|---|
| $\times 4$ | 8.13 | 4.39 | 2.02 |
| $\times 5.7$ | 18.19 | 4.52 | 1.82 |
| $\times 8$ | 34.37 | 4.73 | 1.91 |
| $\times 16$ | 144.97 | 2.75 | 1.81 |
| $\times 32$ | 701.16 | 10.75 | 1.50 |

*Table B.4.* Ablation on model scaling. $E_{spe}$ and $E_{spa}$ denote spatial and spectral encoder depths.

| $(E_{spe}, E_{spa})$ | Params (M) | FLOPs | PSNR(↑) | SAM(↓) | ERGAS(↓) | SSIM(↑) |
|---|---|---|---|---|---|---|
| $(1, 1)$ | 2.735 | 58.873G | $42.45_{\pm 3.24}$ | $5.32_{\pm 1.20}$ | $3.92_{\pm 2.05}$ | $0.985_{\pm 0.004}$ |
| $(2, 2)$ | 9.213 | 0.120T | $44.15_{\pm 3.12}$ | $3.94_{\pm 1.05}$ | $2.88_{\pm 1.50}$ | $0.988_{\pm 0.003}$ |
| $(4, 4)$ | 13.934 | 0.161T | $44.72_{\pm 3.04}$ | $3.05_{\pm 0.95}$ | $2.11_{\pm 1.28}$ | $0.991_{\pm 0.002}$ |
| $(6, 6)$ | 18.654 | 0.202T | $45.96_{\pm 4.69}$ | $2.55_{\pm 1.00}$ | $1.72_{\pm 1.13}$ | $0.994_{\pm 0.005}$ |

*Table B.5.* Leave-one-sensor-out pilot on Botswana (Hyperion, 145 bands, $\times 4$ scale). The reference corresponds to the standard all-sensors joint-training result.

| Protocol | PSNR(↑) | SAM(↓) | ERGAS(↓) | SSIM(↑) |
|---|---|---|---|---|
| Padding+mask (no adaptation) | 42.67 | 1.71 | 2.54 | 0.983 |
| Padding+mask (+500-step adaptation) | 43.59 | 1.47 | 2.09 | 0.985 |
| SSA/MK (no adaptation) | 43.82 | 1.34 | 2.06 | 0.987 |
| SSA/MK (+500-step adaptation) | 45.08 | 0.96 | 1.69 | 0.991 |
| All-sensors joint-training reference | 45.31 | 0.80 | 1.55 | 0.992 |

## B.5. Ablation on Model Scaling

To systematically study how model capacity influences the final reconstruction quality, we conduct an ablation analysis by progressively scaling the depth of the spatial/spectral encoders. These structural changes adjust the total parameter count from 2.735M to 18.65M and FLOPs from 58.873G to 0.202T.

As explicitly shown in Tab. B.4, deploying deeper encoders consistently improves reconstruction accuracy across all metrics; for instance, the PSNR increases by over 3.5 dB from the smallest to the largest configuration. This clear trend indicates that larger models provide significantly stronger spatial–spectral representation capabilities, allowing them to better capture fine-grained features and fit the complex, high-dimensional distributions of hyperspectral data. Although this expansion naturally increases model size and computational cost, such scaling is expected and, as our results suggest, often necessary for data-driven fusion frameworks to achieve state-of-the-art performance when handling large-scale, heterogeneous hyperspectral signals.

## B.6. Ablation: Leave-One-Sensor-Out Protocol

We conduct a leave-one-sensor-out pilot in which the target sensor is entirely excluded from training, as shown in Tab. B.5. We train SSA on the remaining datasets and evaluate on the held-out sensor under the same degradation protocol. Under this stricter setting, the no-adaptation result is 43.82, and after only 500 adaptation iterations, it improves to 45.08, compared with an all-sensors joint-training reference of 45.31. This supports a more careful claim: the learned representation exhibits nontrivial cross-sensor transfer, but the strongest evidence at present is for few-step cross-sensor adaptation, not fully unconditional unseen-sensor generalization.

In the leave-one-sensor-out pilot, the padding-based shared interface shows weaker cross-sensor transfer than MK on a held-out sensor. In particular, MK improves over padding+mask by 1.15 dB without adaptation and by 1.49 dB after 500-step adaptation. This provides direct evidence that MK improves cross-sensor generalization rather than simply enabling compatibility. Mechanistically, zero-padding introduces padded channels with no physical meaning and may expose a fixed zero pattern absent from real spectra. A hard masking implementation is closely related in spirit, but would retain the full kernel shape and suppress invalid channels explicitly, whereas MK directly instantiates the valid kernel subset.

## B.7. More Main Results

To comprehensively evaluate the universality and robustness of our proposed SSA framework, we extend our experiments to five additional benchmarks: Harvard, PaviaU, Chikusei, Botswana, and WashingtonDC. These datasets encompass a wide variety of sensor characteristics, with spectral bands ranging from 31 to 191. As presented in Tabs. B.6 through B.10, our unified model consistently achieves state-of-the-art performance across these diverse scenarios. Remarkably, even though our universal model is trained jointly on heterogeneous datasets, it outperforms specialized approaches like LRTN (Liu et al., 2025) and PSTUN (Wang et al., 2025a) that are trained individually for each specific dataset. For instance, in the in-distribution setting ($\times 4$ scale) on the WashingtonDC dataset (Tab. B.10), our method surpasses the second-best approach by approximately 1.0 dB in PSNR. This result strongly validates that our Matryoshka Kernel design effectively mitigates the interference typically associated with multi-dataset training, allowing a single network to adapt to varying spectral distributions without architectural modification.

We further corroborate these quantitative findings with comprehensive visual comparisons provided in Figs. B.1 through B.6. To explicitly demonstrate the scale generalization capability of our method, we structured the visualizations for the five additional datasets (Figs. B.3–B.6) to display the training scale ($\times 4$, top two rows) and the extreme unseen scale ($\times 32$, bottom two rows) side-by-side. For the CAVE and PaviaC datasets, we specifically showcase the challenging $\times 32$ results in Fig. B.1. As observed, while most competitive methods maintain reasonable quality at the $\times 4$ scale, their performance degrades significantly at $\times 32$. Methods with fixed upsampling modules, such as DCFormer (Wu et al., 2025), completely fail to generate valid outputs at this unseen resolution (indicated by crossed-out boxes), and others exhibit severe blurring or grid-like artifacts. In contrast, leveraging the continuous modeling capability of our INR backbone, SSA maintains high reconstruction fidelity with predominantly dark blue error maps across all scales. Notably, in the Harvard dataset (Fig. B.2), our model successfully recovers legible text (e.g., "30th EDITION") even at the $\times 32$ scale, whereas other methods suffer from heavy distortions.

*Table B.6.* Quantitative comparison on the **Harvard** dataset across four scales. Values are mean $\pm$ std in a single cell.

| | Metric | Harvard Dataset | | | | | | |
|---|---|---|---|---|---|---|---|---|
| | | **DSPNet** | **MIMO-SST** | **DCINN** | **DCFormer** | **PSTUN** | **LRTN** | **Ours** |
| In-Dist. ×4 | PSNR(↑) | 43.97 ± 5.99 | 43.65 ± 6.85 | 42.06 ± 4.55 | 42.87 ± 5.30 | 44.30 ± 4.59 | 41.91 ± 5.33 | 44.81 ± 5.21 |
| | SAM(↓) | 2.70 ± 0.83 | 2.98 ± 0.86 | 2.96 ± 0.85 | 3.19 ± 0.91 | 3.59 ± 0.98 | 4.58 ± 1.54 | 2.73 ± 0.89 |
| | ERGAS(↓) | 4.30 ± 2.29 | 5.11 ± 2.08 | 5.46 ± 0.90 | 5.12 ± 2.46 | 4.54 ± 1.80 | 5.78 ± 1.38 | 4.41 ± 2.63 |
| | SSIM(↑) | 0.983 ± 0.001 | 0.983 ± 0.007 | 0.983 ± 0.003 | 0.982 ± 0.009 | 0.983 ± 0.008 | 0.975 ± 0.008 | 0.984 ± 0.010 |
| Out-of-Dist. ×8 | PSNR(↑) | 42.25 ± 6.20 | 38.62 ± 6.74 | 40.61 ± 6.23 | 42.37 ± 5.25 | 41.79 ± 4.04 | 41.33 ± 5.63 | 42.73 ± 5.09 |
| | SAM(↓) | 2.99 ± 0.90 | 3.26 ± 0.98 | 3.33 ± 0.90 | 3.51 ± 0.93 | 4.00 ± 0.99 | 4.82 ± 1.63 | 3.02 ± 0.92 |
| | ERGAS(↓) | 5.19 ± 2.69 | 6.68 ± 2.43 | 5.69 ± 2.51 | 5.30 ± 2.15 | 5.62 ± 1.84 | 5.98 ± 1.23 | 4.92 ± 2.08 |
| | SSIM(↑) | 0.976 ± 0.012 | 0.944 ± 0.047 | 0.974 ± 0.012 | 0.980 ± 0.012 | 0.977 ± 0.008 | 0.972 ± 0.013 | 0.978 ± 0.013 |
| ×16 | PSNR(↑) | 37.31 ± 6.99 | 34.36 ± 7.28 | 35.94 ± 6.79 | – | 37.46 ± 5.23 | 38.10 ± 6.48 | 38.73 ± 6.00 |
| | SAM(↓) | 3.60 ± 1.15 | 4.01 ± 1.25 | 3.95 ± 1.11 | – | 4.89 ± 1.24 | 5.05 ± 1.56 | 3.85 ± 1.00 |
| | ERGAS(↓) | 7.53 ± 4.48 | 10.74 ± 4.12 | 8.90 ± 2.98 | – | 9.40 ± 5.05 | 8.56 ± 4.39 | 7.68 ± 5.21 |
| | SSIM(↑) | 0.956 ± 0.031 | 0.903 ± 0.069 | 0.950 ± 0.024 | – | 0.951 ± 0.036 | 0.952 ± 0.036 | 0.960 ± 0.037 |
| ×32 | PSNR(↑) | 36.73 ± 6.72 | 31.39 ± 7.65 | 31.78 ± 7.14 | – | 35.78 ± 4.84 | 36.97 ± 6.37 | 37.34 ± 6.49 |
| | SAM(↓) | 4.28 ± 1.38 | 5.63 ± 1.56 | 4.99 ± 1.27 | – | 5.94 ± 1.81 | 5.68 ± 1.91 | 4.66 ± 1.57 |
| | ERGAS(↓) | 9.21 ± 5.44 | 13.41 ± 5.37 | 12.46 ± 4.68 | – | 11.57 ± 5.72 | 9.50 ± 4.14 | 8.70 ± 4.79 |
| | SSIM(↑) | 0.942 ± 0.044 | 0.863 ± 0.090 | 0.902 ± 0.053 | – | 0.941 ± 0.033 | 0.945 ± 0.038 | 0.949 ± 0.045 |

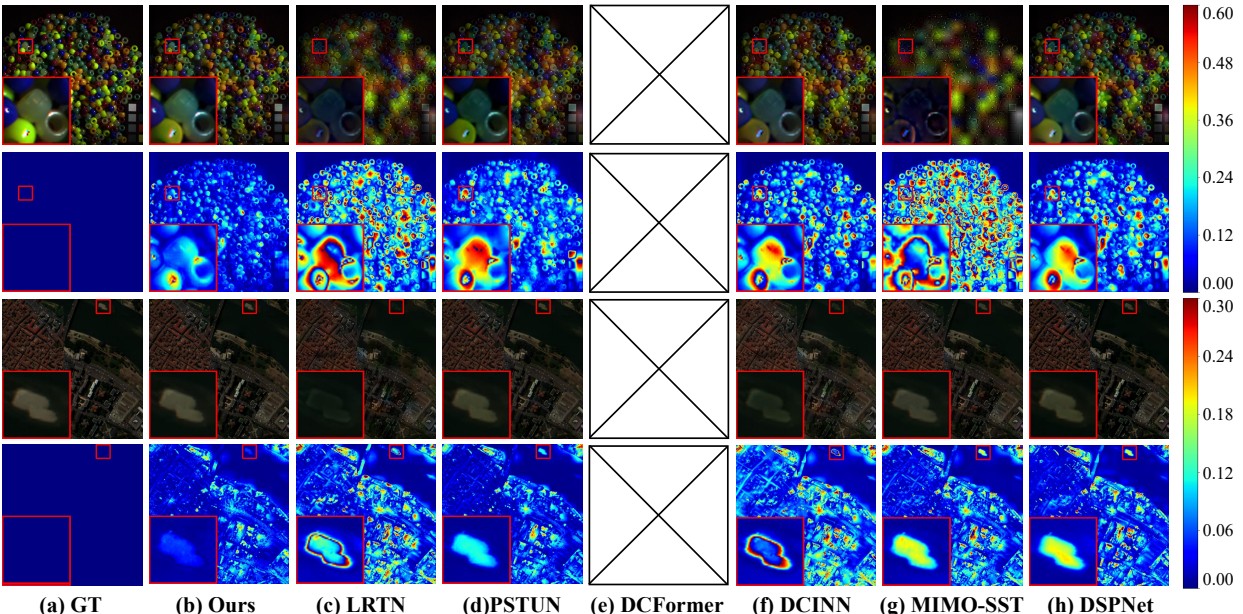

**(a) GT**  **(b) Ours**  **(c) LRTN**  **(d)PSTUN**  **(e) DCFormer**  **(f) DCINN**  **(g) MIMO-SST**  **(h) DSPNet**

*Figure B.1.* Qualitative comparison on the **CAVE** (top rows) and **PaviaC** (bottom rows) datasets at the unseen ×32 **scale**. The red boxes highlight close-up details, and the even rows display the corresponding error maps. Our SSA model exhibits superior detail recovery and minimal reconstruction error compared to SOTA methods, which suffer from significant blurriness at this extreme scale.

*Table B.7.* Quantitative comparison on the **PaviaU** dataset across four scales. Values are mean ± std.

| | | Metric | PaviaU Dataset | | | | | | |
|---|---|---|---|---|---|---|---|---|---|
| | | | DSPNet | MIMO-SST | DCINN | DCFormer | PSTUN | LRTN | Ours |
| In-Dist. | ×4 | PSNR(↑) | 37.51 ± 0.15 | 40.28 ± 0.15 | 40.31 ± 0.42 | 39.05 ± 0.25 | 39.46 ± 0.17 | 40.09 ± 0.63 | 40.44 ± 0.15 |
| | | SAM(↓) | 2.27 ± 0.12 | 3.46 ± 0.11 | 1.96 ± 0.07 | 2.60 ± 0.17 | 2.54 ± 0.02 | 2.05 ± 0.14 | 1.95 ± 0.07 |
| | | ERGAS(↓) | 1.99 ± 0.07 | 1.85 ± 0.11 | 1.53 ± 0.02 | 1.77 ± 0.07 | 1.58 ± 0.02 | 1.55 ± 0.00 | 1.56 ± 0.06 |
| | | SSIM(↑) | 0.981 ± 0.001 | 0.977 ± 0.002 | 0.988 ± 0.001 | 0.977 ± 0.000 | 0.981 ± 0.000 | 0.987 ± 0.001 | 0.988 ± 0.001 |
| Out-of-Dist. | ×8 | PSNR(↑) | 35.76 ± 0.50 | 31.43 ± 0.55 | 28.91 ± 0.44 | 30.56 ± 1.74 | 36.01 ± 0.26 | 36.18 ± 0.03 | 36.68 ± 0.26 |
| | | SAM(↓) | 5.57 ± 0.06 | 5.83 ± 0.17 | 5.36 ± 0.26 | 5.39 ± 0.24 | 4.79 ± 0.27 | 5.16 ± 0.20 | 4.97 ± 0.30 |
| | | ERGAS(↓) | 4.64 ± 0.01 | 4.47 ± 0.12 | 4.89 ± 0.19 | 4.06 ± 0.56 | 3.50 ± 0.15 | 4.01 ± 0.14 | 3.03 ± 0.06 |
| | | SSIM(↑) | 0.917 ± 0.001 | 0.895 ± 0.001 | 0.884 ± 0.003 | 0.927 ± 0.004 | 0.959 ± 0.002 | 0.940 ± 0.001 | 0.960 ± 0.002 |
| | ×16 | PSNR(↑) | 32.76 ± 0.79 | 27.29 ± 1.12 | 26.47 ± 0.56 | – | 33.66 ± 0.49 | 33.26 ± 0.00 | 35.48 ± 0.16 |
| | | SAM(↓) | 7.63 ± 0.31 | 9.18 ± 0.16 | 8.37 ± 0.34 | – | 6.35 ± 0.14 | 6.38 ± 0.23 | 6.26 ± 0.34 |
| | | ERGAS(↓) | 5.72 ± 0.47 | 7.01 ± 0.78 | 6.51 ± 0.25 | – | 5.62 ± 0.01 | 5.03 ± 0.15 | 4.40 ± 0.06 |
| | | SSIM(↑) | 0.873 ± 0.004 | 0.844 ± 0.005 | 0.833 ± 0.005 | – | 0.840 ± 0.003 | 0.879 ± 0.003 | 0.912 ± 0.000 |
| | ×32 | PSNR(↑) | 30.50 ± 1.66 | 24.39 ± 1.25 | 24.25 ± 0.21 | – | 31.49 ± 0.64 | 30.44 ± 0.18 | 34.35 ± 0.14 |
| | | SAM(↓) | 10.50 ± 0.16 | 12.63 ± 0.87 | 11.47 ± 0.49 | – | 9.17 ± 0.08 | 8.81 ± 0.06 | 8.43 ± 0.04 |
| | | ERGAS(↓) | 7.90 ± 1.16 | 9.98 ± 1.69 | 8.38 ± 0.10 | – | 6.60 ± 0.21 | 6.02 ± 0.47 | 5.74 ± 0.13 |
| | | SSIM(↑) | 0.849 ± 0.014 | 0.802 ± 0.018 | 0.809 ± 0.003 | – | 0.829 ± 0.005 | 0.872 ± 0.006 | 0.898 ± 0.000 |

*Table B.8.* Quantitative comparison on the **Chikusei** dataset across four scales. Values are mean ± std.

| | | Metric | Chikusei Dataset | | | | | | |
|---|---|---|---|---|---|---|---|---|---|
| | | | DSPNet | MIMO-SST | DCINN | DCFormer | PSTUN | LRTN | Ours |
| In-Dist. | ×4 | PSNR(↑) | 37.21 ± 3.89 | 36.54 ± 4.12 | 38.05 ± 3.76 | 36.92 ± 3.55 | 38.62 ± 4.26 | 37.88 ± 3.91 | 39.35 ± 4.59 |
| | | SAM(↓) | 2.87 ± 0.62 | 3.05 ± 0.58 | 2.53 ± 0.47 | 2.91 ± 0.54 | 2.22 ± 0.50 | 2.18 ± 0.49 | 2.12 ± 0.46 |
| | | ERGAS(↓) | 5.28 ± 1.47 | 5.63 ± 1.52 | 4.89 ± 1.39 | 5.35 ± 1.41 | 4.54 ± 1.35 | 4.97 ± 1.43 | 4.50 ± 1.42 |
| | | SSIM(↑) | 0.951 ± 0.018 | 0.947 ± 0.021 | 0.958 ± 0.016 | 0.949 ± 0.019 | 0.965 ± 0.014 | 0.976 ± 0.015 | 0.972 ± 0.016 |
| Out-of-Dist. | ×8 | PSNR(↑) | 31.05 ± 2.67 | 30.42 ± 2.79 | 31.58 ± 2.53 | 32.90 ± 2.81 | 30.87 ± 2.48 | 31.22 ± 2.64 | 33.76 ± 1.92 |
| | | SAM(↓) | 4.68 ± 0.91 | 4.92 ± 0.87 | 3.85 ± 0.83 | 3.87 ± 0.84 | 4.75 ± 0.89 | 3.21 ± 0.86 | 3.25 ± 0.78 |
| | | ERGAS(↓) | 7.65 ± 1.02 | 7.98 ± 0.97 | 7.32 ± 0.99 | 6.89 ± 0.94 | 7.72 ± 1.01 | 7.41 ± 0.96 | 6.23 ± 0.51 |
| | | SSIM(↑) | 0.862 ± 0.024 | 0.855 ± 0.027 | 0.871 ± 0.023 | 0.892 ± 0.022 | 0.888 ± 0.021 | 0.867 ± 0.024 | 0.905 ± 0.015 |
| | ×16 | PSNR(↑) | 28.14 ± 2.19 | 27.56 ± 2.33 | 28.62 ± 2.08 | – | 29.29 ± 2.31 | 28.45 ± 2.15 | 30.16 ± 2.94 |
| | | SAM(↓) | 7.24 ± 1.38 | 7.59 ± 1.42 | 6.93 ± 1.31 | – | 6.56 ± 1.25 | 7.08 ± 1.34 | 5.42 ± 1.09 |
| | | ERGAS(↓) | 10.87 ± 0.89 | 11.23 ± 0.95 | 10.54 ± 0.92 | – | 10.09 ± 0.94 | 10.62 ± 0.91 | 8.70 ± 1.07 |
| | | SSIM(↑) | 0.795 ± 0.034 | 0.788 ± 0.037 | 0.802 ± 0.032 | – | 0.818 ± 0.031 | 0.836 ± 0.031 | 0.832 ± 0.033 |
| | ×32 | PSNR(↑) | 26.37 ± 1.05 | 25.89 ± 1.18 | 26.74 ± 1.02 | – | 27.49 ± 0.99 | 26.58 ± 1.07 | 28.46 ± 1.61 |
| | | SAM(↓) | 9.15 ± 1.62 | 9.53 ± 1.71 | 8.87 ± 1.58 | – | 8.49 ± 1.55 | 8.96 ± 1.60 | 6.64 ± 1.23 |
| | | ERGAS(↓) | 12.68 ± 0.81 | 13.05 ± 0.87 | 12.34 ± 0.79 | – | 11.90 ± 0.75 | 12.47 ± 0.82 | 9.92 ± 0.25 |
| | | SSIM(↑) | 0.761 ± 0.011 | 0.753 ± 0.014 | 0.768 ± 0.010 | – | 0.783 ± 0.007 | 0.765 ± 0.012 | 0.803 ± 0.018 |

*Table B.9.* Quantitative comparison on the **Botswana** dataset across four scales. Values are mean $\pm$ std.

| Metric | | | DSPNet | MIMO-SST | DCINN | DCFormer | PSTUN | LRTN | Ours |
|---|---|---|---|---|---|---|---|---|---|
| In-Dist. | ×4 | PSNR(↑) | 42.60 ± 2.32 | 42.97 ± 3.98 | 45.16 ± 3.42 | 41.35 ± 2.81 | 44.94 ± 3.98 | 45.23 ± 4.47 | 45.31 ± 4.30 |
| | | SAM(↓) | 1.42 ± 0.21 | 1.28 ± 0.19 | 0.99 ± 0.11 | 1.47 ± 0.26 | 1.01 ± 0.11 | 0.97 ± 0.10 | 0.80 ± 0.04 |
| | | ERGAS(↓) | 2.01 ± 0.51 | 1.91 ± 0.62 | 1.61 ± 0.49 | 2.14 ± 0.54 | 1.75 ± 0.58 | 1.72 ± 0.60 | 1.55 ± 0.66 |
| | | SSIM(↑) | 0.975 ± 0.005 | 0.980 ± 0.004 | 0.990 ± 0.003 | 0.972 ± 0.005 | 0.989 ± 0.004 | 0.989 ± 0.004 | 0.992 ± 0.004 |
| Out-of-Dist. | ×8 | PSNR(↑) | 38.85 ± 2.38 | 36.60 ± 2.49 | 36.96 ± 2.44 | 39.37 ± 1.72 | 37.21 ± 2.59 | 39.55 ± 2.51 | 39.89 ± 2.36 |
| | | SAM(↓) | 2.15 ± 0.61 | 2.67 ± 1.10 | 2.45 ± 0.93 | 1.96 ± 0.50 | 2.52 ± 0.97 | 2.09 ± 0.82 | 2.13 ± 0.83 |
| | | ERGAS(↓) | 2.87 ± 0.52 | 3.37 ± 0.87 | 2.95 ± 0.67 | 2.60 ± 0.43 | 3.12 ± 0.58 | 2.47 ± 0.52 | 2.49 ± 0.54 |
| | | SSIM(↑) | 0.956 ± 0.004 | 0.926 ± 0.020 | 0.940 ± 0.012 | 0.960 ± 0.003 | 0.946 ± 0.013 | 0.964 ± 0.009 | 0.965 ± 0.008 |
| | ×16 | PSNR(↑) | 36.52 ± 2.39 | 34.15 ± 2.76 | 33.70 ± 2.28 | – | 34.30 ± 2.47 | 36.67 ± 2.26 | 36.85 ± 2.24 |
| | | SAM(↓) | 2.90 ± 1.39 | 3.51 ± 1.67 | 3.26 ± 1.52 | – | 3.40 ± 1.55 | 2.89 ± 1.07 | 2.78 ± 1.27 |
| | | ERGAS(↓) | 3.71 ± 0.81 | 4.34 ± 1.41 | 4.02 ± 1.00 | – | 4.25 ± 0.93 | 3.21 ± 0.73 | 3.28 ± 0.83 |
| | | SSIM(↑) | 0.945 ± 0.009 | 0.906 ± 0.035 | 0.922 ± 0.024 | – | 0.928 ± 0.023 | 0.950 ± 0.017 | 0.951 ± 0.017 |
| | ×32 | PSNR(↑) | 32.60 ± 2.09 | 32.43 ± 2.76 | 32.07 ± 2.27 | – | 31.46 ± 2.44 | 34.72 ± 2.15 | 35.35 ± 2.22 |
| | | SAM(↓) | 4.23 ± 2.08 | 4.30 ± 2.15 | 4.09 ± 2.07 | – | 4.34 ± 1.68 | 3.79 ± 1.67 | 3.64 ± 1.89 |
| | | ERGAS(↓) | 4.89 ± 1.30 | 5.07 ± 1.76 | 4.75 ± 1.42 | – | 4.91 ± 1.16 | 4.23 ± 0.94 | 3.97 ± 1.22 |
| | | SSIM(↑) | 0.918 ± 0.031 | 0.896 ± 0.044 | 0.909 ± 0.034 | – | 0.938 ± 0.025 | 0.939 ± 0.014 | 0.942 ± 0.024 |

*Table B.10.* Quantitative comparison on the **WashingtonDC** dataset across four scales. Values are mean $\pm$ std.

| Metric | | | DSPNet | MIMO-SST | DCINN | DCFormer | PSTUN | LRTN | Ours |
|---|---|---|---|---|---|---|---|---|---|
| In-Dist. | ×4 | PSNR(↑) | 47.64 ± 2.23 | 47.68 ± 2.39 | 47.55 ± 1.78 | 47.18 ± 1.66 | 48.20 ± 1.61 | 47.95 ± 3.07 | 48.91 ± 3.46 |
| | | SAM(↓) | 1.01 ± 0.12 | 1.04 ± 0.16 | 1.04 ± 0.08 | 1.25 ± 0.20 | 1.06 ± 0.16 | 0.86 ± 0.13 | 0.76 ± 0.09 |
| | | ERGAS(↓) | 2.15 ± 0.37 | 2.15 ± 0.44 | 2.24 ± 0.21 | 2.41 ± 0.31 | 1.99 ± 0.28 | 2.48 ± 0.50 | 1.92 ± 0.58 |
| | | SSIM(↑) | 0.995 ± 0.001 | 0.994 ± 0.002 | 0.994 ± 0.000 | 0.994 ± 0.000 | 0.996 ± 0.001 | 0.994 ± 0.001 | 0.996 ± 0.002 |
| Out-of-Dist. | ×8 | PSNR(↑) | 38.65 ± 1.59 | 37.36 ± 1.48 | 40.36 ± 2.18 | 44.74 ± 1.36 | 42.56 ± 2.60 | 41.46 ± 2.60 | 45.08 ± 1.42 |
| | | SAM(↓) | 4.46 ± 0.60 | 5.29 ± 0.82 | 3.24 ± 0.51 | 1.99 ± 0.23 | 2.03 ± 0.37 | 1.86 ± 0.25 | 1.80 ± 0.23 |
| | | ERGAS(↓) | 5.49 ± 0.75 | 6.37 ± 0.72 | 4.86 ± 1.01 | 3.28 ± 0.36 | 3.78 ± 0.73 | 4.44 ± 0.96 | 3.32 ± 0.45 |
| | | SSIM(↑) | 0.957 ± 0.007 | 0.938 ± 0.009 | 0.974 ± 0.008 | 0.989 ± 0.001 | 0.983 ± 0.004 | 0.981 ± 0.005 | 0.990 ± 0.000 |
| | ×16 | PSNR(↑) | 33.73 ± 1.60 | 32.99 ± 1.70 | 35.07 ± 2.12 | – | 39.17 ± 4.13 | 37.32 ± 2.24 | 39.27 ± 2.49 |
| | | SAM(↓) | 7.96 ± 0.78 | 8.69 ± 0.98 | 6.22 ± 0.74 | – | 3.02 ± 0.57 | 3.04 ± 0.42 | 2.97 ± 0.74 |
| | | ERGAS(↓) | 8.94 ± 0.85 | 9.43 ± 1.06 | 8.12 ± 1.09 | – | 5.68 ± 1.68 | 6.44 ± 1.03 | 5.35 ± 1.00 |
| | | SSIM(↑) | 0.912 ± 0.012 | 0.896 ± 0.018 | 0.939 ± 0.014 | – | 0.968 ± 0.025 | 0.961 ± 0.010 | 0.969 ± 0.007 |
| | ×32 | PSNR(↑) | 31.96 ± 1.17 | 31.57 ± 1.42 | 32.45 ± 1.70 | – | 36.54 ± 4.39 | 34.99 ± 1.95 | 37.00 ± 2.02 |
| | | SAM(↓) | 10.35 ± 1.07 | 11.13 ± 1.28 | 8.74 ± 0.96 | – | 3.99 ± 1.17 | 3.94 ± 0.50 | 3.54 ± 0.54 |
| | | ERGAS(↓) | 11.66 ± 0.97 | 11.61 ± 1.05 | 10.99 ± 1.11 | – | 7.13 ± 1.89 | 8.31 ± 1.18 | 6.95 ± 1.31 |
| | | SSIM(↑) | 0.887 ± 0.009 | 0.880 ± 0.010 | 0.910 ± 0.012 | – | 0.951 ± 0.033 | 0.945 ± 0.011 | 0.958 ± 0.009 |

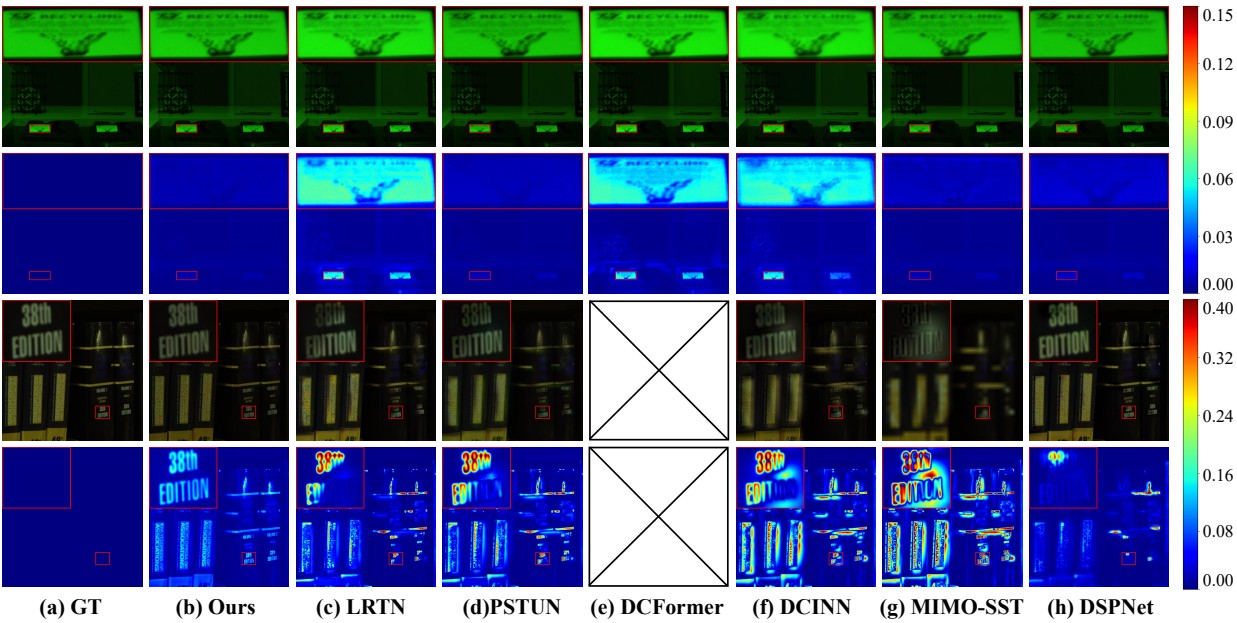

| (a) GT | (b) Ours | (c) LRTN | (d)PSTUN | (e) DCFormer | (f) DCINN | (g) MIMO-SST | (h) DSPNet |

*Figure B.2.* Visual comparison on the **Harvard** dataset. The top two rows correspond to the in-distribution ×4 scale, and the bottom two rows correspond to the unseen ×32 scale. Our method preserves sharp textual details (see zoom-in) even at large magnification factors.

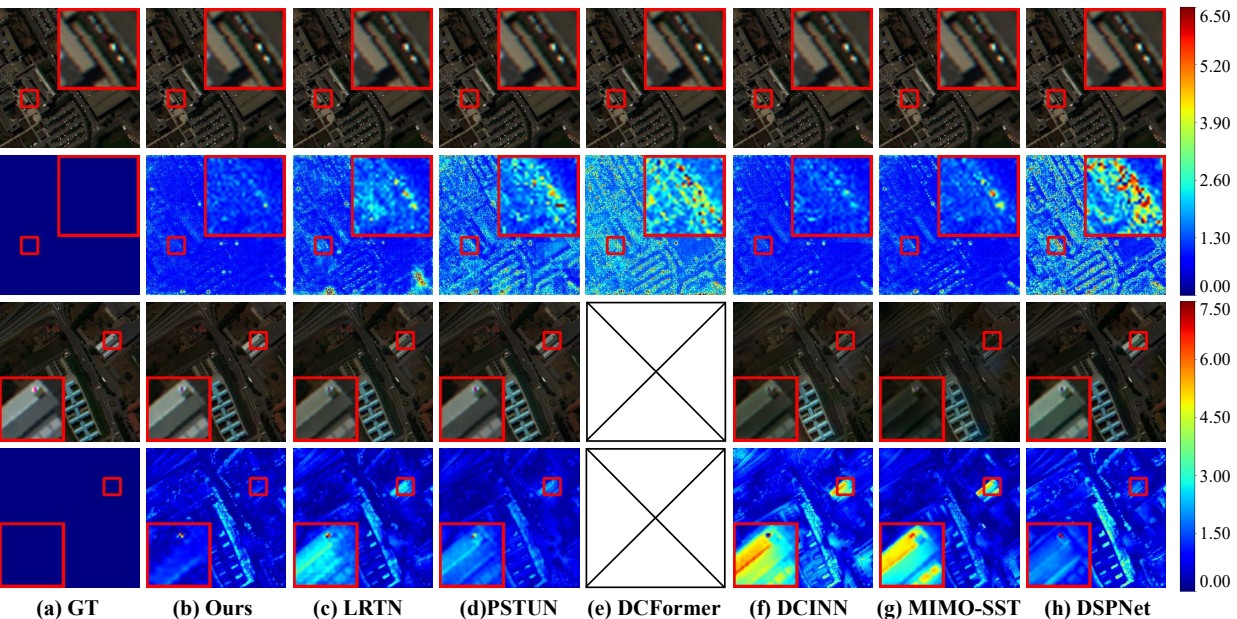

| (a) GT | (b) Ours | (c) LRTN | (d)PSTUN | (e) DCFormer | (f) DCINN | (g) MIMO-SST | (h) DSPNet |

*Figure B.3.* Visual comparison on the **PaviaU** dataset across varying scales. The top two rows show the results at the training scale (×4), while the bottom two rows display the out-of-distribution scale (×32). For each scale, we present the fused image and its corresponding error map. Note that DCFormer (Wu et al., 2025) fails to support the arbitrary ×32 scale (indicated by crossed boxes).

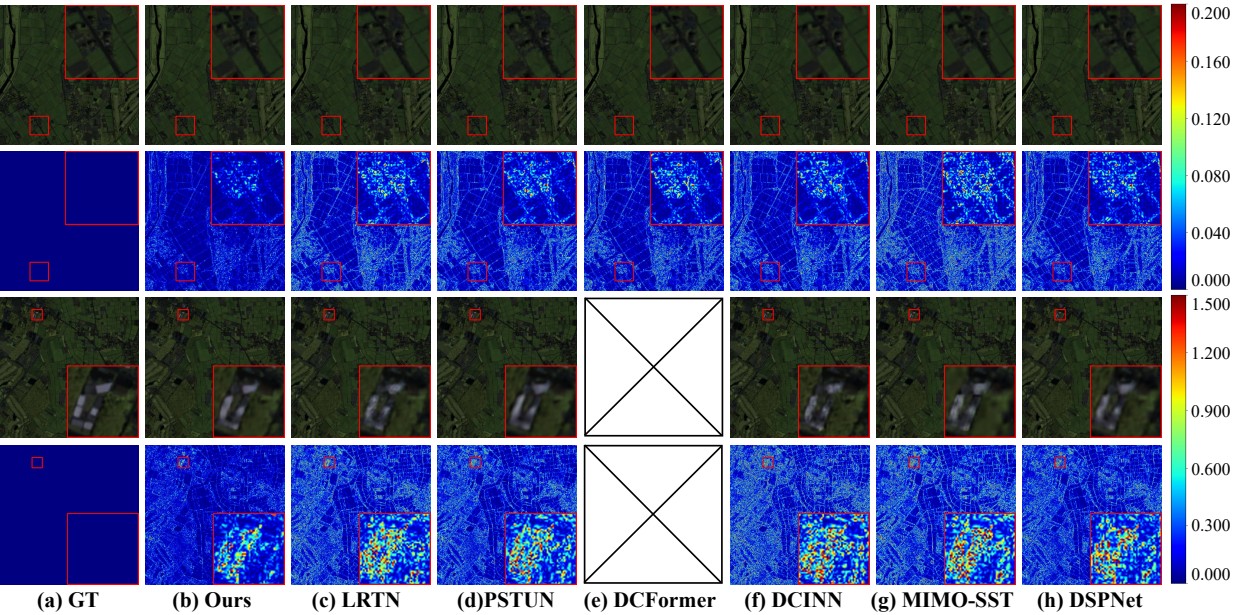

**(a) GT**  **(b) Ours**  **(c) LRTN**  **(d)PSTUN**  **(e) DCFormer**  **(f) DCINN**  **(g) MIMO-SST**  **(h) DSPNet**

*Figure B.4.* Visual comparison on the **Chikusei** dataset. Top two rows: ×4 scale; Bottom two rows: ×32 scale. The error maps (blue indicates low error) demonstrate that our model generalizes well to large-scale remote sensing scenes with varying spatial resolutions.

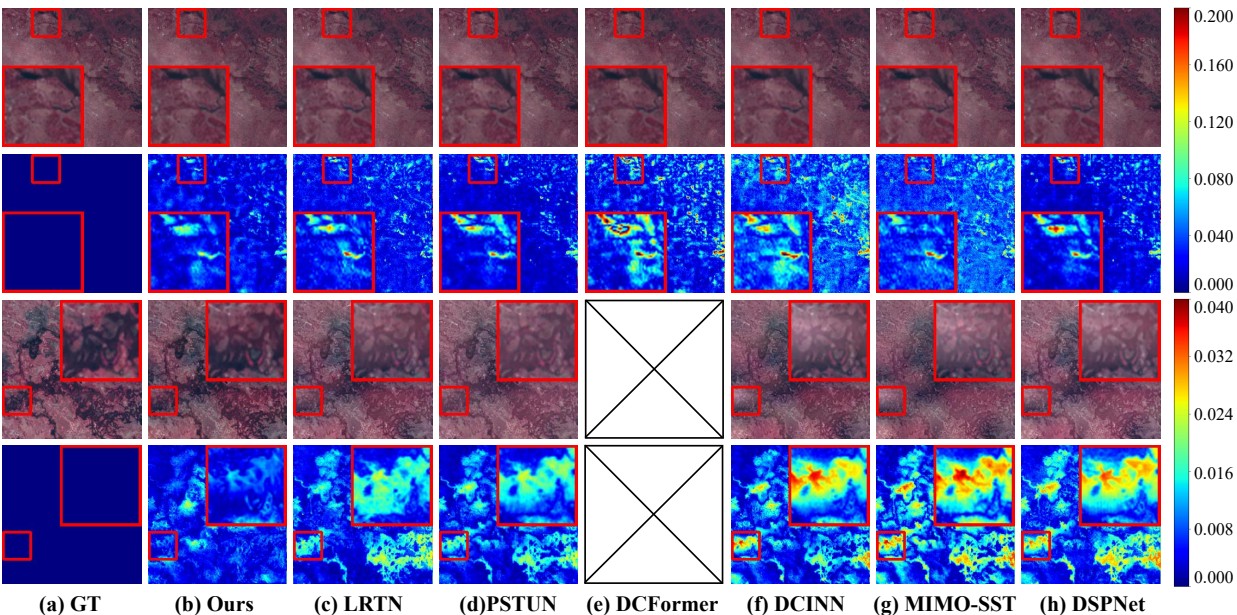

**(a) GT**  **(b) Ours**  **(c) LRTN**  **(d)PSTUN**  **(e) DCFormer**  **(f) DCINN**  **(g) MIMO-SST**  **(h) DSPNet**

*Figure B.5.* Visual comparison on the **Botswana** dataset. Top two rows: ×4 scale; Bottom two rows: ×32 scale. Our approach maintains spectral and spatial consistency better than comparison methods, which often show high-energy residuals (red/yellow) at large upscaling factors.

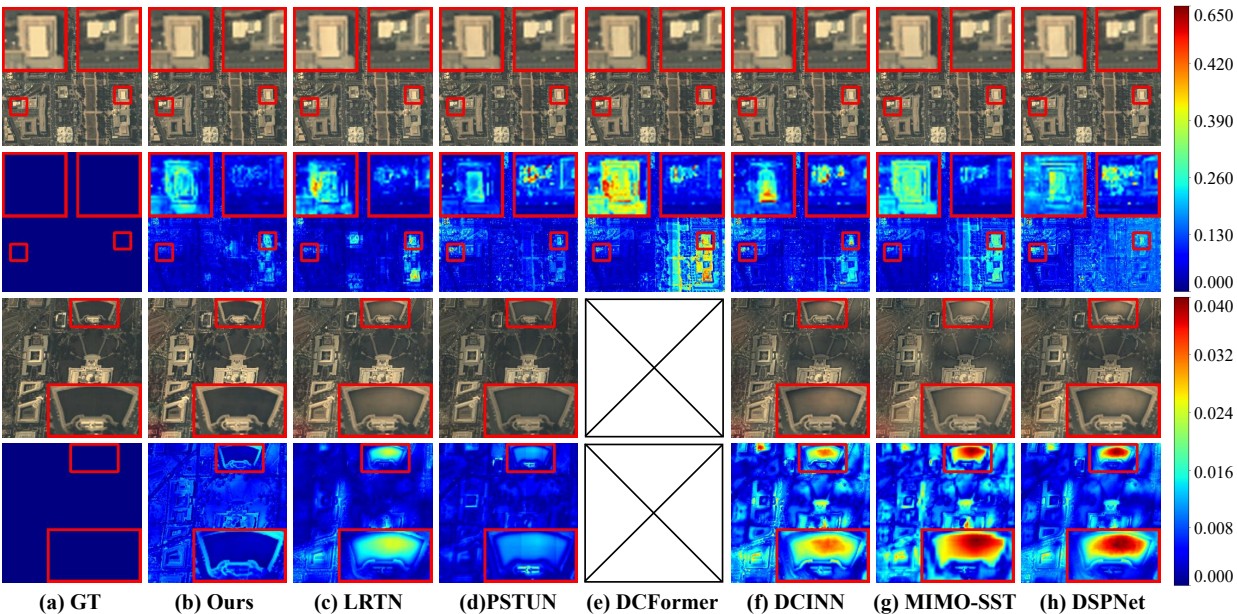

(a) GT    (b) Ours    (c) LRTN    (d)PSTUN    (e) DCFormer    (f) DCINN    (g) MIMO-SST    (h) DSPNet

*Figure B.6.* Visual comparison on the **WashingtonDC** dataset. Top two rows: ×4 scale; Bottom two rows: ×32 scale. Despite the high band count (191 bands), our unified model robustly reconstructs spatial details across different scaling factors without model retraining.

