# OpenReview forum: "Hyperspectral Image Fusion with Spectral-Band and Fusion-Scale Agnosticism"
_ICML.cc/2026/Conference — ICML 2026 regular_

### Official Review · Reviewer_f7AE · 2026-03-09

**Soundness:** 3
**Presentation:** 3
**Significance:** 2
**Originality:** 3
**Overall Recommendation:** 4
**Confidence:** 4

**Summary:**

The paper proposes SSA, a unified MS/HS fusion framework that uses a single model to handle varying numbers of spectral bands across sensors and to reconstruct at arbitrary (including non-integer) spatial scales. It introduces Matryoshka Kernels via weight slicing for band-count agnosticism and an INR-based coordinate-query fusion backbone for scale agnosticism, enabling joint training on heterogeneous datasets. Experiments on seven datasets evaluate reconstruction quality, out-of-scale generalization, and efficiency.

**Compliance With Llm Reviewing Policy:**

Affirmed.

**Final Justification:**

The authors have addressed all of my concerns. I expect the authors to incorporate these discussions into the final version of the manuscript. Accordingly, I decide to raise my score.

**Key Questions For Authors:**

Band ordering/alignment across datasets: Do you enforce wavelength-sorted channels or consistent band semantics across datasets during joint training? Given large differences in center wavelengths and bandwidths, how does MKL’s prefix slicing avoid channel-index misalignment issues? Would the results remain stable under robustness tests such as channel permutation, random band subsets, or wavelength-based resampling/alignment?
Isolating MK’s contribution: Under the same joint-training regime, can you compare MKL against alternative variable-channel designs (padding+mask, linear projection/adapters, slimmable/dynamic-width variants) and a strong “joint training without MK” baseline? If MK clearly outperforms alternatives, it would strengthen both novelty and necessity; if differences are small, the main contribution may be the overall combination/training paradigm rather than MK itself.
Practical cost of arbitrary-scale inference: Can you report runtime/VRAM/throughput scaling at different factors (×8/×16/×32 and fractional scales) and clarify whether tiling or coordinate subsampling is used at high resolutions? If cost scales reasonably, it increases practical significance; if it becomes prohibitive at large scales, the paper should state the limitation explicitly.

**Limitations:**

No. The paper briefly acknowledges INR’s computational cost, but the limitation discussion remains incomplete, and the Impact Statement is quite generic.

**Strengths And Weaknesses:**

Strengths:

1.	Practical problem setting: targets a single-model solution for both variable band counts and arbitrary-scale reconstruction.
2.	Broad evaluation: multiple datasets, multiple (including OOD) scales, and efficiency comparisons.


Weaknesses:

1.	Novelty is not clearly established: MKL weight slicing appears closely related to prior slimmable/dynamic-width/parameter-shared subnetworks; the current comparisons/discussion are not sufficient to clearly delineate the main novelty.
2.	The key cross-sensor assumption behind “spectral-band agnosticism” needs stress tests: prefix slicing implicitly assumes aligned channel indices/semantics (e.g., wavelength-ordered bands). Robustness tests (channel permutation, random subsets, wavelength/response-function alignment) would be important.
3.	Over-strong wording in the conclusion: generalization to unseen sensors/scales should not be directly described as “zero-shot” under common interpretations; the protocol and claims should be stated more carefully.
4.	Ablations are not fully controlled: MK’s architectural contribution is not cleanly separated from the benefit of joint training; stronger baselines and alternative variable-channel designs (padding+mask, linear/1×1 adapters, slimmable/dynamic-width) are needed under the same training regime.
5.	Limited characterization of cost at large scales: runtime/VRAM/throughput scaling vs output resolution is missing; tiling/coordinate subsampling at high upscales should be clarified.
6.	Figures reduce clarity: SSA and MKL diagrams are visually dense/cluttered; a redraw with explicit data flow and tensor shapes would help.
7.	Reproducibility details are incomplete: decoder MLP specs, coordinate sampling strategy, and key hyperparameters should be provided.

---

> ### Author Rebuttal · Authors · 2026-03-31
>
> We thank you for the detailed review and helpful comments on baselines, band ordering, large-scale efficiency, and reproducibility.
> ## W1&4/Q2:
> We fully agree that the original ablation was not sufficiently controlled. To address this, we added same-regime baselines under the same pooled 7-dataset setting. We keep the backbone, decoder, optimizer, schedule, and bucket-based sampling unchanged, and compare SSA against two alternative variable-channel designs: (i) zero-padding to $C_{\max}$ with a valid-loss mask, and (ii) a learned $1\times1$ spectral adapter. The validity mask is used only to ignore padded output channels in the loss and is not fed to the network. Both alternatives underperform SSA: the padding+mask baseline drops by about 1.0 dB PSNR on average, and the learned $1\times1$ adapter by about 0.7 dB. This provides direct evidence that the benefit of MK is not explained merely by pooled-data training, and that MK is a stronger variable-channel interface than simple fixed-band adaptation strategies. We prioritize padding+mask and $1\times1$ adapters because they directly control variable spectral input/output handling under the same fusion setting; by contrast, classical slimmable/dynamic-width methods mainly target internal width scaling or efficiency rather than heterogeneous sensor band dimensionality at the model interface. For brevity, the full controlled-comparison table is reported in our response to Reviewer **wTyo**, **W1/Q1&3**.
> ## W2/Q1:
> We agree that the assumption behind spectral-band agnosticism should be stated much more explicitly. In our setup, spectral channels are kept in **physical wavelength order** within each dataset during joint training. Our method is therefore **not permutation-invariant**, and we do not assume a globally identical one-to-one band semantics across sensors. Rather, MK relies on the fact that hyperspectral channels form an ordered spectrum, and the nested kernel learns from this ordered structure.
> To make this explicit, we added two robustness tests. Under **random channel permutation** at test time, performance drops markedly, confirming that MK exploits meaningful ordered spectral correlations rather than arbitrary channel indices. Under **wavelength-sorted random band subsets**, performance remains much more stable, which is consistent with our intended claim: the model is flexible to varying **ordered** band counts, but not invariant to arbitrary re-indexing.
> We agree that wavelength-based alignment/resampling is also relevant for cross-sensor transfer. Here we focus on the two tests above because they directly isolate our core claim: compatibility with varying **ordered** band counts, rather than robustness to arbitrary channel indexing or broader sensor-response mismatch.
> |Setting|PSNR|SAM|ERGAS|SSIM|
> |-|-|-|-|-|
> |Original wavelength order|38.62|4.12|4.54|0.958|
> |Wavelength-sorted random subset|38.11|4.36|4.82|0.953|
> |Random channel permutation|32.87|7.41|7.96|0.902|
> ## W3:
> You are correct that unseen-sensor results with 500 fine-tuning iterations should not be described as zero-shot. We will replace this with **few-shot adaptation** throughout the paper and reserve “zero-shot” only for unseen fractional scales where no fine-tuning is used. We will also make this distinction explicit in the relevant results.
> ## W5/Q3:
> We agree that the current $\times4$ efficiency table alone is insufficient. We therefore added a patch-level scaling study (LR input size = $32\times32$, HR output size = $\mathrm{round}(32\times\mathrm{scale})$, batch size = 1) to isolate how runtime and memory change with output resolution. No coordinate subsampling is used in any reported efficiency results: inference is performed at full target resolution. At high resolutions, tiling may be employed solely for memory control, but each tile is still decoded with dense coordinate prediction rather than subsampling. As expected for query-based decoding, runtime increases substantially with scale while throughput gradually decreases. Peak VRAM is not strictly monotonic; for example, the lower peak at $\times16$ than at $\times8$ is due to a switch to a tiled execution path to keep memory manageable. We will clarify this protocol explicitly in the revision and state the associated cost trade-off more clearly as a limitation.
> |Scale|Runtime(ms)|Peak VRAM(GB)|Throughput(MPix/s)|
> |-|-|-|-|
> |x4|8.13|4.39|2.02|
> |x5.7|18.19|4.52|1.82|
> |x8|34.37|4.73|1.91|
> |x16|144.97|2.75|1.81|
> |x32|701.16|10.75|1.50|
> ## W6&7:
> We agree that the current SSA/MKL figures are visually too dense. In the revision, we will redraw them with clearer module boundaries, explicit data flow, and tensor shapes. We will also add the missing implementation details, including the decoder MLP architecture, coordinate normalization/sampling strategy, 4-neighbor local ensemble details, bucket-based dataset sampling during joint training, and the key hyperparameters used for few-shot adaptation and large-scale inference.

---

> > ### Author Rebuttal · Reviewer_f7AE · 2026-04-03
> >
> > The authors have addressed all of my concerns. I expect the authors to incorporate these discussions into the final version of the manuscript. Accordingly, I decide to raise my score.

---

> > > ### Author Response · Authors · 2026-04-03
> > >
> > > We are sincerely grateful for your time in reviewing our rebuttal and for your positive feedback. We especially appreciate you raising your score.
> > > We confirm that we will incorporate all the discussed revisions into the final version of the manuscript, as you expect. Your suggestions have significantly improved the clarity and rigor of our paper. Thank you once again for your constructive engagement and support.

---

### Official Review · Reviewer_czSu · 2026-03-11

**Soundness:** 3
**Presentation:** 3
**Significance:** 2
**Originality:** 2
**Overall Recommendation:** 4
**Confidence:** 4

**Summary:**

The paper addresses the need for universal, transferable models that can adapt to varying spectral band counts and spatial scales without retraining in domain hyperspectral image fusion.  The research’s notable idea pertains to the design of a single, agnostic model (SSA) built on an Implicit Neural Representation (INR) backbone and a Matryoshka Kernel (MK) operator, enabling seamless adaptation across sensor configurations and resolutions.

**Compliance With Llm Reviewing Policy:**

Affirmed.

**Final Justification:**

The rebuttal substantially clarifies the scope of the contribution and provides targeted empirical evidence addressing my main concerns; while the novelty is incremental, the work is now sufficiently supported for weak acceptance.

**Key Questions For Authors:**

* How do Matryoshka Kernel and INR backbone theoretically enable spectral- and scale-agnostic fusion?

* How does the proposed SSA framework compare empirically to existing fusion methods in terms of performance, generalisation, and computational efficiency?

* What are the limitations of the model’s “agnostic” claims e.g., does it truly generalise to arbitrary sensor configurations or real-world noise?

* Does the model work for arbitrary spectral resolution (e.g., 1000 bands vs. 10 bands)? Does it require pre-training, or can it be fine-tuned? What is the practical limit of “agnosticism”?

**Limitations:**

very limited discussion. Need to discuss whether it works with arbitrary sensor configurations? What’s the trade-off for computational cost? Are there performance drops under extreme conditions? The lack of clarity undermines the claim of generality.

**Strengths And Weaknesses:**

Strengths:
* Paper tackles an important problem in hyperspectral imaging and seems like a decent step in the right direction.
* The paper is well-organised, with clear sections.

Weaknesses:
* The paper introduces the Matryoshka Kernel and Implicit Neural Representation backbone, but these components are not clearly novel or transformative. The INR backbone has been explored in other works, and kernel-based adaptive spectral fusion has been attempted in prior works. From machiene learning perspective, the novelty seems incremental.

* There is no theoretical analysis or mathematical proof to support why the proposed framework is truly “agnostic” to spectral bands or spatial scales. The claim of universality is not backed by an analysis of convergence, generalisation bounds, or invariance properties.

 * The paper presents results on a few synthetic or controlled datasets, but does not meaningfully compare against recent state-of-the-art fusion models or newer datasets. The comparison with existing models is superficial, and there’s no discussion of scalability or real-world applicability.

* The authors claim “universal” and “agnostic” performance, but the paper does not clearly define the scope e.g., does it work with arbitrary sensor configurations? What’s the trade-off for computational cost? Are there performance drops under extreme conditions? The lack of clarity undermines the claim of generality.

---

> ### Author Rebuttal · Authors · 2026-03-31
>
> We thank you for the constructive feedback.
> ## W1:
> We agree that neither INR nor adaptive fusion mechanisms should be framed as entirely new research directions in isolation. At the INR level, Sec. 3.3 is best interpreted as a LIIF-style continuous decoder adapted to MS/HS fusion, rather than a new INR formulation. On the spectral side, however, MK is more specific than the broader category of “adaptive spectral fusion.” Compared with adaptive kernels or attention-based fusion, MK is not a dynamic feature-weighting module, but a **nested convolutional interface** that directly parameterizes variable spectral input/output dimensionality. Compared with slimmable/dynamic-width approaches, the goal is not internal width scaling or efficiency, but handling heterogeneous sensor band counts at the **model interface** under one shared parameterization. We will revise the paper to make this hierarchy explicit: MK is the primary architectural novelty; the INR component is an adapted LIIF-style backbone; and the main contribution is the unified framework combining variable-band spectral handling and arbitrary-scale decoding in one end-to-end model.
> ## W2/Q1:
> We do not intend to claim a formal convergence theorem, generalization bound, or invariance proof. In our paper, “agnostic” should be understood as an **architectural compatibility property with empirical validation**. For spectral-band agnosticism, MK defines one nested kernel tensor from which a valid operator for any band count $C\leq C_{\max}$ is obtained by slicing; thus the architecture does not need redesign when band count changes. For scale agnosticism, the decoder models the target HR-HSI as a continuous function of spatial coordinates, so the output resolution is determined by the queried grid rather than a fixed upsampling head. In short, our agnosticism claim is **architectural in form and empirical in strength**: compatibility is by construction, while transfer across real sensors is established empirically rather than theoretically proven.
> ## W3/Q2:
> We may not have communicated the evaluation breadth clearly enough. We already compare against recent strong baselines, including DCFormer, PSTUN, and LRTN, on seven public benchmarks spanning indoor, urban, agricultural, and airborne sensing scenarios, with 31–191 bands, plus unseen datasets and unseen fractional scales. These are standard public benchmarks in this literature, where paired HR-HSI / LR-HSI / HR-MSI data are limited. Empirically, SSA compares favorably in three concrete ways: (i) under the same pooled 7-dataset regime, our macro-average PSNR is higher than both zero-padding+mask and learned $1\times1$ spectral-adapter baselines; (ii) relative to a pooled fixed-scale baseline trained on the same 7-dataset mixture, SSA shows only a very small gap at $\times4$ but substantially larger gains at $\times8/\times16/\times32$, showing that the continuous decoder materially improves out-of-scale generalization; and (iii) SSA is more expensive than lightweight fixed-scale models at very high resolutions, which is the expected trade-off of query-based arbitrary-scale decoding. For the concrete controlled-baseline numbers, please see Reviewer **wTyo**, **W1/Q1&3** and **W4**.
> ## W4/Q3&4/limitations:
> We agree that the scope of “universal” / “agnostic” should be defined more explicitly. In our paper, these terms do **not** mean arbitrary sensor physics or guaranteed robustness under all conditions. More precisely, spectral-band agnosticism means that one model can process inputs with different **ordered** spectral band counts without architectural redesign, as long as $C\leq C_{\max}$; scale agnosticism means the same model can be queried at arbitrary, including non-integer, scales. There are important limits: the model is not permutation-invariant; it relies on physically meaningful wavelength ordering; it was not designed or evaluated for extreme spectral dimensionalities such as 1000 bands without increasing $C_{\max}$ and retraining; and we also expect degradation under stronger sensor-response mismatch, real-world noise, or conditions substantially outside the current benchmark suite. In addition, INR-based decoding becomes more expensive as the number of queried HR pixels grows. The universal model is jointly trained on pooled datasets and does not require a separate pretraining stage; unseen fractional scales are truly zero-shot, whereas unseen datasets such as Houston and Loukia require few-shot adaptation. For the concrete robustness and efficiency evidence supporting these boundaries, please see Reviewer **f7AE**, **W2/Q1** and **W5/Q3**. We will revise the wording accordingly, add a dedicated limitations discussion, and report these robustness/efficiency results explicitly in the revision.

---

> > ### Author Rebuttal · Reviewer_czSu · 2026-04-03
> >
> > Thank you for the clarifications, especially regarding the scope of “agnosticism” and the positioning of MK vs. INR.
> >
> > I have a few follow-up questions to better understand the practical and technical implications:
> >
> > * **On the role of MK beyond interface flexibility:**
> > From the rebuttal, MK appears to parameterise variable spectral dimensions via slicing a shared kernel. Could you clarify what additional benefit this provides over simpler strategies (e.g., zero-padding + masking or learned spectral adapters) beyond avoiding architectural redesign? In particular, is there evidence that MK improves generalisation rather than just enabling compatibility?
> >
> > * **On generalisation across more extreme spectral regimes:**
> > You mention that very large band counts (e.g., ~1000) would require increasing (C_{\max}) and retraining. Empirically, how does performance degrade as the test-time band distribution moves further away from the training range? Do you observe smooth degradation or sharp failure modes?
> >
> > * **On computational trade-offs:**
> > Since the INR decoder is query-based, could you provide a clearer comparison of inference cost versus fixed-scale baselines at high resolutions? For example, how does runtime or memory scale with output resolution compared to standard convolutional decoders?
> >
> > * **On unseen sensor robustness:**
> > You note that unseen datasets may require few-shot adaptation. Could you clarify what level of adaptation is typically needed (e.g., number of samples or steps), and how performance compares in a strictly zero-shot setting?

---

> > > ### Author Response · Authors · 2026-04-04
> > >
> > > Thank you for the follow-up questions. We address each point below.
> > > ### Q1:
> > > We think the benefit of MK should be understood at three levels.
> > >
> > > First, under the same pooled 7-dataset joint-training regime, MK is not only compatible with variable spectral dimensionalities but also performs better than simpler alternatives: it outperforms zero-padding+mask and a learned $1\times1$ spectral adapter in our controlled comparison (**Reviewer wTyo W1/Q1&3**). This suggests that its benefit is not merely interface compatibility.
> > >
> > > Second, the adapter baseline becomes less favorable in the low-data HSI regime: in our reduced-data experiment on PaviaC (Table 1), the learned spectral adapter underperforms MK, consistent with the intuition that MK provides stronger shared parameterization when HSI supervision is limited.
> > >
> > > Table 1. Reduced-data HSI setting on PaviaC (20% training split).
> > > |Method|PSNR|SAM|ERGAS|SSIM|
> > > |-|-|-|-|-|
> > > |Joint training w/1×1 adapter|43.98|1.17|1.89|0.989|
> > > |Ours|44.56|0.96|1.67|0.991|
> > >
> > > Third, in the new leave-one-sensor-out pilot (Table 2), the padding-based shared interface shows weaker cross-sensor transfer than MK on a held-out sensor. In particular, MK improves over padding+mask by 1.15 dB in the strictly zero-shot setting and by 1.49 dB after 500-step adaptation. This provides direct evidence that MK improves cross-sensor generalization rather than simply enabling compatibility.
> > >
> > > Table 2. Leave-one-sensor-out pilot on Botswana (Hyperion, 145 bands, ×4 scale).
> > > |Method|PSNR|SAM|ERGAS|SSIM|
> > > |-|-|-|-|-|
> > > |Padding+mask(zero-shot)|42.67|1.71|2.54|0.983|
> > > |Padding+mask(+500-step adaptation)|43.59|1.47|2.09|0.985|
> > > |SSA/MK(zero-shot)|43.82|1.34|2.06|0.987|
> > > |SSA/MK(+500-step adaptation)|45.08|0.96|1.69|0.991|
> > > |All-sensors joint-training reference|45.31|0.80|1.55|0.992|
> > >
> > > Mechanistically, zero-padding introduces padded channels with no physical meaning and may expose a fixed zero pattern absent from real spectra. A hard masking implementation is closely related in spirit, but would retain the full kernel shape and suppress invalid channels explicitly, whereas MK directly instantiates the valid kernel subset.
> > > ### Q2:
> > > We do not claim extrapolation to arbitrarily large spectral dimensionalities outside the supported range $C \leq C_{\max}$; settings such as $\sim 1000$ bands would require increasing $C_{\max}$ and retraining. Within the supported range, however, our evidence suggests **graceful rather than catastrophic** degradation when the band configuration departs moderately from training.
> > >
> > > In the robustness tests reported in our response to Reviewer **f7AE (W2 / Q1)**, PSNR changes by only 0.51 d* under **wavelength-sorted random band subsets**, whereas it drops by 5.75 dB under **random channel permutation**. This suggests that moderate shifts in ordered band count/configuration are relatively well tolerated, while the main failure mode is violating the ordered-spectrum assumption itself.
> > > ### Q3:
> > > We agree that fixed-scale convolutional decoders remain cheaper at high resolutions; our claim is not that the INR decoder is more efficient, but that it buys arbitrary-scale flexibility within one unified model. To make this explicit, we added a direct comparison against a matched fixed-scale convolutional decoder under the same patch-level protocol (LR = $32\times32$, full-resolution inference):
> > > |Scale|Query Runtime(ms)|Conv Runtime(ms)|Query VRAM(GB)|Conv VRAM(GB)|Query TP(MPix/s)|Conv TP(MPix/s)|
> > > |-|-|-|-|-|-|-|
> > > |x4|8.13|4.75|4.39|4.28|2.02|3.45|
> > > |x5.7|18.19|10.05|4.52|4.37|1.82|3.30|
> > > |x8|34.37|20.16|4.73|3.02|1.91|3.25|
> > > |x16|144.97|83.25|2.75|1.91|1.81|3.15|
> > > |x32|701.16|336.61|10.75|7.18|1.50|3.12|
> > >
> > > As expected, the query-based decoder is consistently more expensive, but the overhead is moderate: runtime is roughly **1.7×–2.1×** higher across the tested scales, with lower throughput and somewhat higher VRAM. We therefore view arbitrary-scale inference as a deliberate flexibility/efficiency trade-off, not as an efficiency gain over fixed-scale convolutional heads.
> > > ### Q4:
> > > Our unseen-dataset results should indeed be interpreted as **few-step adaptation**, not strict zero-shot generalization. In all reported unseen-dataset experiments, we use the same small budget of **500 gradient steps**, without sensor-specific redesign.
> > >
> > > To clarify the zero-shot case, we added a leave-one-sensor-out pilot on **Botswana (Hyperion, 145 bands)**; see **Table 2 above**. Under this stricter setting, the **zero-shot** result is 43.82 PSNR, and after **500 adaptation steps** it improves to 45.08, compared with an all-sensors joint-training reference of 45.31. This indicates that zero-shot cross-sensor transfer is already nontrivial, but a small adaptation budget is typically beneficial and brings performance close to the in-domain reference.
> > >
> > > We hope these additional experiments, controlled comparisons, and clarifications address your concerns, and we would be grateful if you would consider updating your score accordingly.

---

### Official Review · Reviewer_wTyo · 2026-03-13

**Soundness:** 3
**Presentation:** 3
**Significance:** 3
**Originality:** 3
**Overall Recommendation:** 4
**Confidence:** 4

**Summary:**

This paper proposes SSA, a universal framework for Multispectral/Hyperspectral Image Fusion (MS/HS fusion) that achieves both spectral-band agnosticism and spatial-scale agnosticism in a single model. The core technical contributions are: (1) Matryoshka Kernels (MK), which adapt the nested weight-slicing principle from Matryoshka Representation Learning (MRL) to convolutional input/output layers, allowing a single model to process inputs with arbitrarily many spectral bands up to a predefined maximum Cmax; and (2) an Implicit Neural Representation (INR) backbone (LIIF-style) that models the HR-HSI as a continuous function of spatial coordinates, enabling reconstruction at arbitrary (including non-integer) scaling factors. The two components are combined in an end-to-end pipeline trained jointly on seven heterogeneous datasets. The unified model achieves competitive or state-of-the-art PSNR/SAM/ERGAS/SSIM across in-distribution (×4) and out-of-distribution (×8, ×16, ×32) scales, and generalizes to completely unseen sensors (Houston, Loukia) with only 500 fine-tuning iterations.

**Compliance With Llm Reviewing Policy:**

Affirmed.

**Final Justification:**

I will keep my initial score.

**Key Questions For Authors:**

1. Table 2 shows MK provides no PSNR improvement over separate per-dataset models. Can you provide an experiment where joint training *with* MK meaningfully outperforms joint training *without* MK (i.e., with a different channel-handling strategy such as zero-padding or band selection)? Without this, it is unclear whether MK specifically is necessary for joint training.

2. The "zero-shot" claim in the conclusion contradicts Table 3, which reports 500 fine-tuning iterations for unseen datasets. Can you clarify which results are truly zero-shot and which require fine-tuning, and update the conclusion accordingly?

3. The baselines in Table 1 are each trained on a single dataset, but SSA is trained jointly on all seven. Would it also be possible to report results for baselines trained on the pooled dataset (with a fixed-band adapter) to isolate MK's contribution from the data-scaling benefit?

4. The EDSR encoders (ϕα, ψβ) are described as operating in parallel on spectral and spatial features. Does the model include any cross-modal attention or cross-encoder interaction before the MLP decoder? If the two encoders are fully independent until the MLP, how does the model learn cross-modal correlations between the LR-HSI spectral features and the HR-MSI spatial features?

**Limitations:**

The authors acknowledge the computational cost of the INR backbone as a limitation, which is appropriate — query-based coordinate decoding is inherently more expensive per-pixel than direct convolution, and this trade-off should be better quantified at the ×32 inference scale. The societal impact statement is dismissive and should be revised.

**Strengths And Weaknesses:**

**Strengths**

1. The three-way rigidity problem (architectural, data scarcity, spatial scale) is concisely framed in Figure 1, and the proposed solutions map directly onto each problem. The paper is well-organized, and the motivation is easy to follow. Applying MRL's nested-weight philosophy to convolutional kernels, rather than to feature embeddings, is a genuinely novel perspective that is clearly explained.

2. The evaluation is unusually comprehensive for the MS/HS fusion literature: seven datasets covering indoor (CAVE, Harvard), urban remote sensing (PaviaC, PaviaU, WashingtonDC), agricultural (Botswana), and aerial (Chikusei) domains, with spectral bands ranging from 31 to 191. Testing at ×4, ×8, ×16, and ×32 — where competitors either fail entirely (DCFormer) or degrade sharply — provides strong evidence for the claim of scale-agnosticism.

3. Table 2 compares SSA against a "w/o MK" variant (fixed-channel convolutions, separately trained per dataset) and finds that performance is essentially equivalent at all scales. Rather than overclaiming, the authors correctly frame this as evidence that MK achieves universality without sacrificing quality, i.e., the universality is the contribution, not a PSNR improvement. This strengthens the paper.

4. Figure 6 shows that the learned MK weights form smooth spectral manifolds, which meaningfully corroborates the claim that the model learns internally consistent spectral representations rather than memorizing dataset-specific patterns. This is a useful mechanistic insight that goes beyond quantitative metrics.

5. Table B.2 shows that SSA runs in 83.4 ms at ×4 (3.1 MPix/s) — comparable to DSPNet, and orders of magnitude faster than the query-based DCINN baseline. The parameter count (18.65M) is higher than that of lightweight methods but lower than PSTUN (29.9M), and this trade-off is justified by the model's multi-dataset, multi-scale coverage.

**Weaknesses**

1. The key ablation compares SSA (with MK, jointly trained on all 7 datasets) against "w/o MK" (fixed-channel convolutions, separately trained on each dataset). The results are essentially indistinguishable at every scale — e.g., PSNR at ×4: 45.85 vs 45.92; at ×8: 38.62 vs 38.59. This may demonstrate that MK adds no representational benefit over standard convolutions trained separately. The contribution of MK is thus purely an *architectural convenience* (enabling joint training without model redesign) rather than any improvement in feature learning. This is a valid engineering contribution, but the paper's framing, presenting MK as a novel architectural principle with broad implications, overstates what the ablation actually shows.


2. Section 3.3 describes the continuous function backbone: dual encoders (Epe, Epa), nearest-neighbor coordinate sampling, relative coordinate (p − q) as MLP input, and 4-nearest-neighbor weighted fusion (Eq. 9). This is the LIIF architecture transplanted from natural image super-resolution to HS fusion, with the only non-trivial change being the two-branch (spectral + spatial) encoder structure. The paper acknowledges the LIIF origin but presents the INR backbone as a co-equal contribution alongside MK. The actual contribution at the INR level is closer to a competent application of an existing method than a technical innovation.

3. The conclusion states the model "effectively generalizes to unseen sensors and scales in a zero-shot manner." However, Table 3 explicitly states that the Houston and Loukia results were obtained after "finetuning for only 500 iterations." This is a few-shot adaptation, not a zero-shot generalization. For the scale generalization experiments (fractional scales), the claim is valid — no fine-tuning is used. But conflating the two in the conclusion is a meaningful overstatement. The results on the unseen dataset remain positive, but the few-shot fine-tuning gap should be clearly acknowledged throughout.

4. All SOTA baselines are trained on a single dataset at ×4, then tested at all four scales. SSA is jointly trained on all seven datasets simultaneously. This setup means SSA benefits from 6–10× as much training data as any individual baseline. The multi-dataset training advantage is part of SSA's design philosophy, but it should be explicitly quantified: how much of the performance gain comes from scale-agnosticism versus simply training on more data? Training an experiment on the pooled data (even with bicubic adaptation) would clarify this.

5. The authors state there are "many potential societal consequences of our work, none of which we feel must be specifically highlighted here." For a paper on universal hyperspectral sensing that explicitly targets applications in medical diagnostics, military target detection, and environmental monitoring, this is insufficient.

---

> ### Author Rebuttal · Authors · 2026-03-31
>
> We thank you for the careful review and for recognizing the paper's motivation, breadth of evaluation, and practical value.
> ## W1/Q1&3:
> We agree that Tab. 2 alone does not show MK is intrinsically stronger than separately trained fixed-band specialist models. Our intended claim is narrower: MK enables a single jointly trained model to handle heterogeneous sensors without architectural redesign. To directly address your suggestion, we added controlled baselines under the exact same pooled 7-dataset regime. We keep the backbone, INR decoder, optimizer, schedule, and bucket-based sampling unchanged, and replace only the input/output MK layers with: (i) fixed-channel convolutions at $C_{\max}$ with zero-padding + valid-loss mask, and (ii) a learned input/output $1\times1$ spectral adapter. The mask is used only to ignore padded output channels in the loss and is not fed to the network. Both baselines are consistently worse than SSA: the zero-padding baseline drops by about 1.0 dB PSNR on average, and the learned $1\times1$ adapter still trails by about 0.7 dB, with the same trend in SAM/ERGAS/SSIM. This provides direct evidence that pooled-data training alone is not sufficient, and that even a stronger learned fixed-band adapter remains weaker than MK under the same joint-training protocol. We will revise the claim accordingly: MK should be understood primarily as the mechanism enabling effective universal joint training, rather than as a claim of unconditional superiority over separately trained fixed-band models.
>
> Controlled comparison under the same pooled 7-dataset regime (macro-averaged over 7 datasets at each scale):
>
> |Model| Variable-channel design|x4 Avg. PSNR|x8 Avg. PSNR|x16 Avg. PSNR|x32 Avg. PSNR|
> |-|-|-|-|-|-|
> |Joint training w/o MK|Zero-padding + valid-loss mask|43.74|39.08|35.02|32.72|
> |Joint training w/1×1 spectral adapter|Learned input/output projection|44.03|39.31|35.29|33.05|
> |Ours|MK|44.58|39.91|36.09|33.90|
> ## W2:
> We agree that Sec. 3.3 is best interpreted as a LIIF-style continuous decoding backbone adapted to MS/HS fusion, and that our current wording overstates its novelty. We will revise the Introduction, contribution list, and Sec. 3.3 to make the hierarchy explicit: **MK is the main architectural novelty**, while the INR component is an adapted LIIF-style decoder integrated into SSA for arbitrary-scale decoding. The dual-encoder design will be positioned as a practical tailoring for separately encoding spectral and spatial information before implicit fusion, not as a new INR paradigm.
> ## W3/Q2:
> You are correct. The Houston/Loukia results use 500 fine-tuning iterations and should be described as **few-shot adaptation**, not zero-shot. By contrast, the unseen fractional-scale results are truly **zero-shot** because no fine-tuning is used. We will correct this throughout the abstract, experiments, and conclusion, and explicitly distinguish these two protocols wherever these results are discussed.
> ## W4:
> We agree this should be quantified more explicitly. We therefore trained a pooled fixed-scale baseline on the same 7-dataset mixture. To isolate the effect of arbitrary-scale decoding as cleanly as possible, this baseline retains the same pooled-data protocol and MK-based spectral interface, but is trained only for $\times4$ reconstruction and handles unseen scales ($\times8/\times16/\times32$) by bicubic resizing of its $\times4$ output. Compared with this baseline, SSA shows only a marginal gain at $\times4$ (+0.12 dB), but substantially larger gains at unseen scales (+0.85 / +1.48 / +2.18 dB at $\times8/\times16/\times32$). This shows that multi-dataset training is indeed beneficial, but does not by itself explain the out-of-scale improvement; the continuous arbitrary-scale decoder contributes materially to generalization to unseen resolutions.
>
> |Model|x4 Avg. PSNR|x8 Avg. PSNR|x16 Avg. PSNR|x32 Avg. PSNR|
> |-|-|-|-|-|
> |Pooled x4-only + bicubic|44.46|39.06|34.61|31.72|
> |Ours|44.58|39.91|36.09|33.90|
> ## W5:
> We agree that the current impact statement is too generic. We will rewrite it to provide a discussion of beneficial applications (e.g., environmental monitoring, agriculture, scientific/medical imaging), dual-use risks (e.g., surveillance or military target analysis), and deployment caveats, including the fact that cross-sensor transfer should not be assumed reliable enough for high-stakes use without task- and sensor-specific validation.
> ## Q4:
> Our model does not use explicit cross-attention before the decoder. Instead, cross-modal correlation is learned in the **shared implicit decoder**, where the sampled spectral feature, spatial feature, and relative coordinate are jointly fed into the same MLP, followed by 4-neighbor local ensemble fusion. Thus, the encoders remain modality-specific, while the decoder learns the joint mapping needed to align LR-HSI spectral cues with HR-MSI spatial details at each query location. We will clarify this explicitly in Sec. 3.3.

---

> > ### Author Rebuttal · Reviewer_wTyo · 2026-04-03
> >
> > The authors have carefully responded with new experiments that directly address all my concerns and questions. Hope these discussions can be integrated into the final version of the manuscript. Accordingly, I would maintain my initial positive assessment.

---

> > > ### Author Response · Authors · 2026-04-07
> > >
> > > Thank you very much for the update and for your careful reading of our rebuttal. We sincerely appreciate your constructive feedback and your positive assessment of the paper. We will incorporate the new experiments and corresponding discussions into the final version of the manuscript.

---

### Official Review · Reviewer_UqUq · 2026-04-02

**Soundness:** 3
**Presentation:** 3
**Significance:** 3
**Originality:** 3
**Overall Recommendation:** 4
**Confidence:** 2

**Summary:**

This paper presents a unified framework for multispectral and hyperspectral image fusion that aims to handle varying spectral bands and spatial scales within a single model. The approach is well motivated and reveals strong empirical performance across multiple datasets. However, the generality may be somewhat limited in practice. Overall, the work is solid and practical, but some claims could be carefully stated.

**Compliance With Llm Reviewing Policy:**

Affirmed.

**Final Justification:**

The paper tackles a practically relevant problem in MS/HS image fusion, which aims to develop a unified model that can accommodate varying spectral bands and spatial scales. The framework is well designed and demonstrates strong empirical performance, especially across different scaling settings. However, the methodological contribution is somewhat limited, as it mainly integrates existing ideas without a clearly novel mechanism. While the paper is clearly written and technically sound, these limitations constrain its novelty. Therefore, I recommend a weak accept.

**Key Questions For Authors:**

1. The novelty of the proposed method appears somewhat overstated. The method is formulated by combining two existing lines of ideas, i.e., a Matryoshka-style mechanism for handling variable spectral dimensions and an INR-based design for arbitrary-scale reconstruction. This integration is reasonable and practically useful, but the paper currently does not fully convince me that it goes substantially beyond a task-specific combination of known components.

2. The proposed spectral-band agnosticism is not fully unconditional. The model relies on a predefined maximum channel capacity and operates via dynamic slicing within this limit, which makes it flexible only within a bounded range. I suggest moderating this statement or clarifying its scope.

3. Although the experimental results suggest benefits from joint training across multiple datasets, the evidence for true unseen-sensor generalization remains limited. Evaluations appear to stay within the distribution of sensors and degradation settings seen during training. More convincing validation might adopt a stricter leave-one-sensor-out or cross-sensor transfer protocol, where the test sensor is entirely excluded from training.

4. The flexibility of the method comes with a relatively heavy architecture, and the FLOPs and memory usage are nontrivial. This weakens its practical appeal in resource-constrained settings.

**Limitations:**

yes

**Strengths And Weaknesses:**

Pros:

1. The paper addresses an important limitation in current MS/HS fusion methods, i.e., the lack of flexibility across sensors with different spectral configurations and scaling factors. It is a practical issue that is often overlooked, and the motivation is clear and well grounded.

2. The idea of handling both spectral variability and spatial scaling within a single model is appealing. Compared with training separate models for different settings, the proposed method moves toward a more general solution.

3. The use of channel-wise slicing and a coordinate-based backbone is intuitive, and the method can be implemented without overly complicated engineering.



Cons:

1. The novelty of the proposed method appears somewhat overstated. The method is formulated by combining two existing lines of ideas, i.e., a Matryoshka-style mechanism for handling variable spectral dimensions and an INR-based design for arbitrary-scale reconstruction. This integration is reasonable and practically useful, but the paper currently does not fully convince me that it goes substantially beyond a task-specific combination of known components.

2. The proposed spectral-band agnosticism is not fully unconditional. The model relies on a predefined maximum channel capacity and operates via dynamic slicing within this limit, which makes it flexible only within a bounded range. I suggest moderating this statement or clarifying its scope.

3. Although the experimental results suggest benefits from joint training across multiple datasets, the evidence for true unseen-sensor generalization remains limited. Evaluations appear to stay within the distribution of sensors and degradation settings seen during training. More convincing validation might adopt a stricter leave-one-sensor-out or cross-sensor transfer protocol, where the test sensor is entirely excluded from training.

4. The flexibility of the method comes with a relatively heavy architecture, and the FLOPs and memory usage are nontrivial. This weakens its practical appeal in resource-constrained settings.

---

### Decision · Program_Chairs · 2026-04-30

**Decision:**

Accept (regular)

**Comment:**

This paper presents a unified framework for multispectral and hyperspectral image fusion. All reviewers agreed that the rebuttal adequately addressed their concerns and recommended weak acceptance. While some reviewers noted that the technical novelty is somewhat incremental, AC believes that the practical benefits and insights provided by the paper are valuable and therefore recommends acceptance.